# THE EVER-EVOLVING SCIENCE EXAM

## ABSTRACT

As foundation models grow rapidly in capability and deployment, evaluating their scientific understanding becomes increasingly critical. Existing science benchmarks have made progress towards broad **Range**, wide **Reach**, and high **Rigor**, yet they often face two major challenges: **data leakage risks** that compromise benchmarking validity, and **evaluation inefficiency** due to large-scale testing. To address these issues, we introduce the **Ever-Evolving Science Exam (EESE)**, a dynamic benchmark designed to reliably assess scientific capabilities in foundation models. Our approach consists of two components: 1) a non-public **EESE-Pool** with over 100K expertly constructed science instances (question-answer pairs) across 5 disciplines and 500+ subfields, built through a multi-stage pipeline ensuring Range, Reach, and Rigor, 2) a periodically updated 500-instance subset **EESE**, sampled and validated to enable leakage-resilient, low-overhead evaluations. Experiments on 32 open- and closed-source models demonstrate that EESE effectively differentiates the strengths and weaknesses of models in scientific fields and cognitive dimensions. Overall, EESE provides a robust, scalable, and forward-compatible solution for science benchmark design, offering a realistic measure of how well foundation models handle science questions.

## 1 INTRODUCTION

With the rapid development of large-scale foundation models, there arises an urgent need to evaluate their scientific abilities in a reliable and systematic way (Zhang et al., 2025b; Bommasani et al., 2021; Ouyang et al., 2022; Wang et al., 2025c; Firoozi et al., 2025). Science benchmarks play a vital role in this process, offering a standardized, quantitative foundation for assessing how well models understand and reason about scientific concepts. As science benchmarks continue to evolve, the research community is gradually converging on a shared understanding of what defines a high-quality science benchmark (e.g., MMLU (Hendrycks et al., 2020), SuperGPQA (Du et al., 2025), GSM8K (Cobbe et al., 2021), ScienceQA (Lu et al., 2022), HLE (Phan et al., 2025), SciEval (Sun et al., 2024)). Naturally, this prompts the question:

*What constitutes a good science benchmark?*

In general, an ideal benchmark should meet three essential criteria: broad **Range**, wide **Reach**, and high **Rigor**, which together ensure that it is: *1) Extensive in scale* (Range): comprising a large volume of instances to support robust and statistically meaningful evaluation, *2) Diverse in scope* (Reach): spanning a broad array of scientific disciplines and offering varied question formats to capture different cognitive and reasoning skills, *3) Sound in methodology* (Rigor): constructed through a careful, principled pipeline with rigorous quality assurance and verification processes.

While many existing benchmarks strive to meet these criteria, new challenges emerge that limit their effectiveness in evaluating the scientific capacities of foundation models. First, there is a growing concern about **data leakage** (Xu et al., 2024; Zhou et al., 2025b; López et al., 2024; Wu et al., 2024). Once a benchmark is publicly available, there is a non-negligible risk that it could be inadvertently included in training data, especially when data is gathered via large-scale web scraping. Such leakage distorts the evaluation valid-

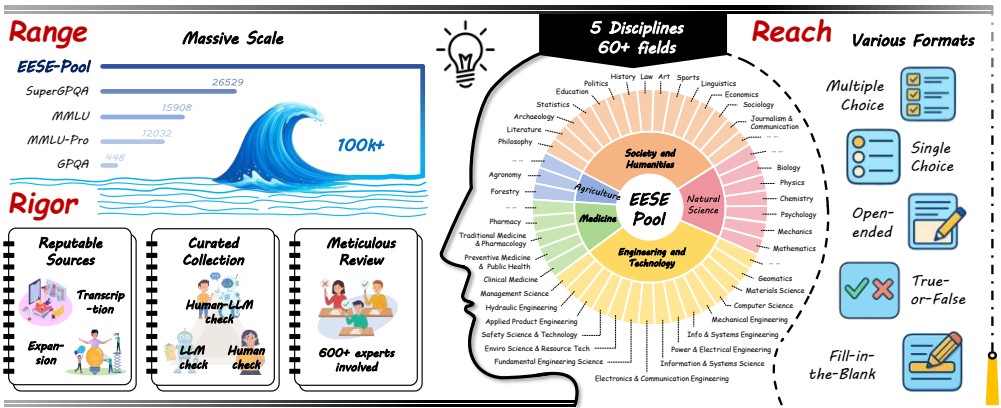

Figure 1: Overview of EESE-Pool construction, which adheres to the principles of **Range** (vast quantity of instances), **Reach** (diverse field and question format), and **Rigor** (systematic and rigor data construction). Specifically, EESE-Pool comprises over 100K science question–answer pairs spanning 5 disciplines and over 500 subfields.

ity, making performance scores unreliable. Second, there is the issue of **evaluation inefficiency** (Zhou et al., 2025a; Zhang et al., 2025c; Gupta et al., 2024; Wen et al., 2025). While increasing the number of evaluation instances can improve benchmark reliability, large-scale evaluation introduces significant computational and financial overheads. This evaluation cost can hinder rapid iteration in model development.

To balance high-quality benchmark design with practical needs like leakage-resistance and evaluation efficiency, we propose a new benchmark: **The Ever-Evolving Science Exam (EESE)**. Concretely, a two-level strategy is adopted: 1) We build a large-scale, high-quality, non-public instances repository, named EESE-Pool, which contains over 100,000 science instances. This pool is constructed under strict principles of **Range**, **Reach**, and **Rigor**. 2) We periodically sample a dynamic subset of 500 instances, called EESE, for actual evaluation. This subset is carefully curated to maintain Range, Reach, and Rigor, while mitigating **leakage risk** and reducing **evaluation inefficiency** through regular updates. Hence, EESE not only faithful and aligned with the principles of a good science benchmark, but offers low-cost, leakage-resistant, and continuously refreshed evaluations that better reflect real-world generalization and robustness of model.

To construct EESE-Pool, we design a streamlined **Data Engine** that ensures Range, Reach, and Rigor through three stages. In the *Transcription* stage, we collect raw instances from textbooks, public databases, and online sources. These instances are then standardized into a unified format and classified into 163 subfields based on academic taxonomy (Press, 2009). In the *Expansion* stage, these initial fields are enriched by engaging experts to develop high-quality instances, expanding the coverage to over 500 subfields. In the *Categorization* stage, we assign difficulty levels to each instance by evaluating model performance and manually validating correctness. To raise instance quality and mitigate trivial or ambiguous cases, a dedicated **Data Refinement** process is introduced. This process strategically improves the instance through a *Parallel Three-Branch Refinement Framework*: Enhancement By Distraction, Enrichment By Cross-Disciplinary, and Refinement By Expert.

To derive EESE, a representative, regular-updating, leakage-resilient, and low-overhead, evaluation set, we adopt a dynamic sampling strategy alongside expert check on EESE-Pool. Notably, we evaluate 32 leading models on EESE-Pool and EESE, and provide actionable guidance for the development of forward-compatible science benchmarks. In summary, our key contributions are as follows:

- **A large-scale, high-quality science benchmark pool:** We construct EESE-Pool, a 100K+ science question-answer pair pool across 5 disciplines and 500+ subfields, with diverse formats and rigorous quality control. We design three-stage Data Engine (Transcription, Expansion, and Categorization) and Data Refinement (a Parallel Three-Branch Refinement Framework) to ensure range, reach, and rigor.

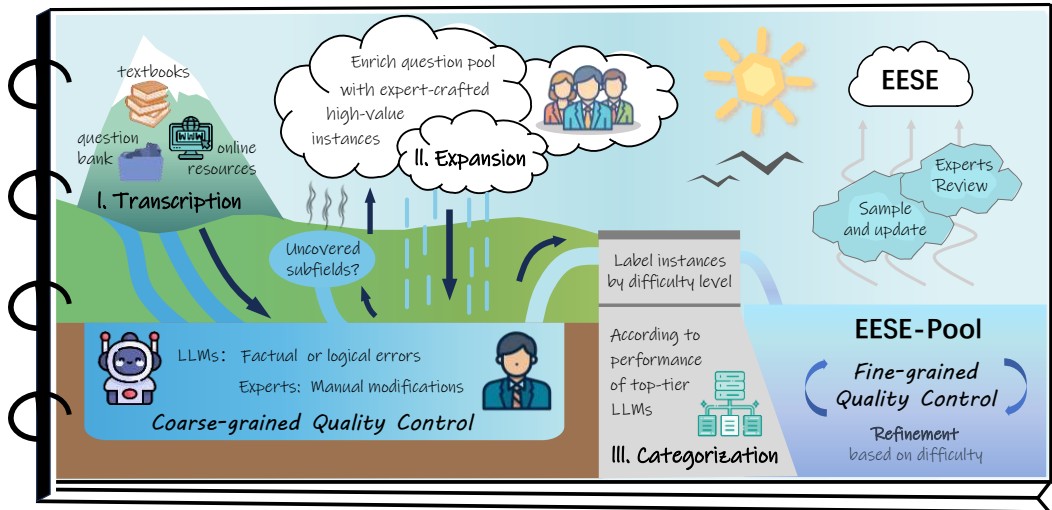

Figure 2: EESE-Pool Construction Framework. The three-stage **Data Engine** (Transcription, Expansion, Categorization) with a systematic **Data Refinement** process ensures large-scale coverage, expert-enriched content, difficulty stratification, and iterative quality improvement, laying a foundation for dynamic, leakage-resilient EESE.

- **A dynamic, leakage-resilient evaluation set:** We propose EESE, a 500-instance subset periodically updated (regular resampling 500 instances from the EESE-Pool), maintaining representativeness while reducing leakage risk and evaluation overhead.
- **Comprehensive evaluation of LLMs:** We evaluate 32 leading models (open- and closed-source) on EESE-Pool and EESE, revealing significant performance gaps across disciplines, the effectiveness of refinement in improving quality, and the trade-offs between inference cost and science ability. The findings offer insights for future science benchmarks.

## 2 PRINCIPLES

An ideal science benchmark is expected to embody large scale, broad disciplinary, format diversity, and methodological robustness. In alignment with these expectations, **EESE-Pool** is founded upon the principles of **Range**, **Reach**, and **Rigor**. As illustrated in Figure 1, these three principles together define EESE-Pool as a reliable question pool for evaluating scientific capabilities in foundation models:

*I. Range → The vast quantity of science instances within EESE-Pool.*

We construct EESE-Pool as a dynamic and expansive question pool, containing over 100,000 carefully collected instances (question-answer pairs). These instances are collected from a wide spectrum of scientific disciplines, ensuring that the pool covers a broad and representative **Range**.

This Range significantly exceeds most existing science benchmarks, supporting the long-term stability of the evaluation system and laying a solid foundation for diverse instance selection. Building on this comprehensive Range, we construct EESE, *a regularly updated subset of 500 instances*. The breadth of EESE-Pool ensures that EESE remains representative across field, difficulty levels, and cognitive dimensions.

*II. Reach → The coverage of EESE-Pool across disciplines and question formats.*

EESE-Pool spans five disciplines and over 500 subfields based on standard academic taxonomy (Press, 2009). It also supports a wide range of question formats, including single-choice, multiple-choice, fill-in-the-blank, true/false, and open-ended questions.

**Increasing Human Involvement**

| Low ~ Distraction |
| --- |
| ***Original Question:*** *A protocol suite is ( ). A. A set of protocols. B. A hierarchical collection of protocols.* |
| ***After Refinement:*** *Regarding protocol suites, which of the statements is correct? ( ). A. A given protocol suite can only run on one type of computer. B. Each layer adds a header to packets received from higher layers of the protocol suite. C. A protocol suite is a hierarchical collection of protocols. D. Each layer provides services to the next higher layer.* |

Medium ~ Cross-Disciplinary

***Original Question:*** *Given an element with a maximum oxidation state of +7, determine its period and group.*

***After Refinement:*** *Elements A, B, C, and D are from period 4. A forms a 1:1 compound with D. B is a d-block element with oxidation state +7. C is in the same period and has the same oxidation state as B. D is the most electronegative element. Fill in the table below and order the four elements by electronegativity from high to low .*

| | Element | Symbol | Period | Group | Max Oxidation |
| --- | --- | --- | --- | --- | --- |
| A | | | | | |
| B | | | | | |
| C | | | | | |
| D | | | | | |

High ~ Expert-Driven

***Original Question:*** *A machine has a 16-bit instruction word with a 6-bit address field. If the opcode is 4 bits long, how many 0-address instructions are possible?*

***After Refinement:*** *A machine uses 16-bit instruction words and 6-bit operand addresses. Assume the opcode length is fixed, with instructions in three formats: 0-, 1-, and 2-address. Given M 0-address and N 1-address instructions, what's the maximum number of 2-address instructions? If opcode length is variable, what's the maximum number of 2-address instructions?*

Figure 3: Data refinement of EESE-Pool. Candidate instances are systematically improved through three refinement paths: *Enhancement by Distraction*, *Enrichment by Cross-Disciplinary*, and *Expert-Driven Refinement*. This multi-level human involvement strategy effectively raises instance difficulty, ensuring robust and discriminative evaluation.

This broad field **Reach** includes both natural and social sciences, enhancing the evaluation of reasoning and social cognition. The diverse formats enable the benchmark to assess a wide spectrum of capabilities, from knowledge retrieval to complex reasoning.

***III. Rigor → The systematic and principled processes that ensure quality in EESE-Pool and EESE.***

EESE-Pool undergoes a **Rigor** construction process that incorporates both coarse- and fine-grained quality control. Coarse-grained control is implemented via the **Data Engine**, while fine-grained control is achieved through **Data Refinement** using a three-branch refinement pathway strategy. EESE is then randomly sampled and further manually modified by field experts.

This rigorous construction process ensures that EESE-Pool maintains consistent quality standards, and that EESE reliably reflects the intended challenge level.

## 3 THE EESE

### 3.1 DATA ENGINE

To construct the EESE-Pool with broad **Range**, wide **Reach**, and high **Rigor**, we build a streamlined Data Engine pipeline, as shown in Figure 2. This pipeline comprises three sequential stages: *Transcription*, *Expansion*, and *Categorization*, described in detail below.

***I. Transcription → Raw data from diverse sources is collected and uniformly transcribed***.

**Transcription** is collecting and standardizing raw data into a unified format, forming the foundation of EESE-Pool. Transcription represents a widely adopted, efficient methodology for rapid large-scale benchmark construction (Zhong et al., 2023; Hendrycks et al., 2020; Huang et al., 2023; Chen et al., 2025). To implement this, over 300 experts from academic institutions collect instances from textbooks, question banks, and online resources, transcribing them into a standardized format. Notably, a two-step coarse-grained quality control measure is employed: 1) Experts deploy a suite of powerful LLMs to flag instances with errors in formatting, factual accuracy, or logical coherence. 2) Experts review and manual modify the flagged instances. Subsequently, the transcribed instances are categorized into 163 subfields according to the standard disciplinary taxonomy (Press, 2009), and classified by format including multiple-choice, multiple-answer, fill-in-the-blank, true/false, and open-ended questions.

***II. Expansion → Enrich question pool with expert-crafted instances for specific fields***.

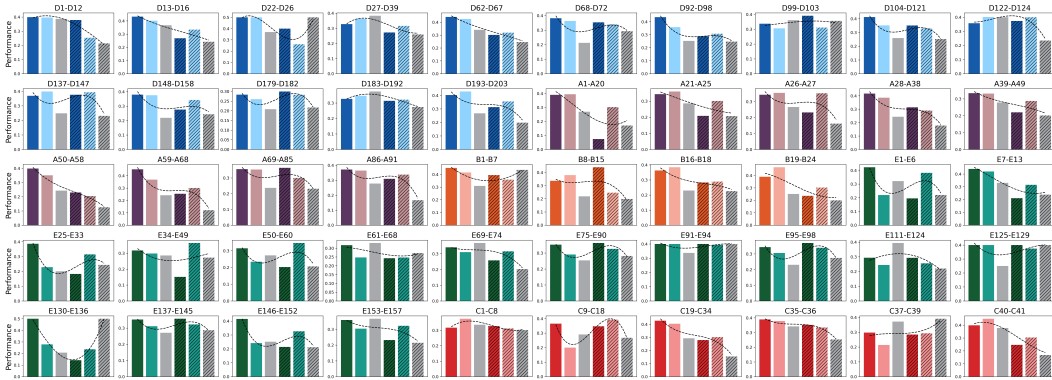

Figure 4: Performance of six leading models evaluated on the EESE-Pool, leveraging over 100K expertly verified instances and comprising more than 600k model inferences (evaluated across 50 representative fields). Each subplot corresponds to a field by its label (such as 'D1-D12', see appendix) and is color-coded by its parent discipline: ETS (blue), NS (purple), AS (orange), SSH (green), and MS (red). Bars from left to right in each subplot represent the average performance for O3, Gemini-2.5-Pro, GPT-4o, DeepSeek-R1, Qwen-2.5-72B-Instruct, and Grok-3.

**Expansion** is systematically extending the benchmark to over 500 subfields, addressing initial field coverage gaps while enforcing strict quality control. For predefined subfields that are currently uncovered or insufficiently represented, experienced specialists are responsible for contributing high-value instances. These instances are developed through the synthesis of field knowledge, practical experience, and pedagogical insights. To address potential deviations in human-crafted answers, all instances undergo rigorous verification (coarse-grained quality control) to ensure consistency and reliability. This stage ensures **comprehensive coverage of over 500 subfields** while guaranteeing the quality of EESE-Pool.

*III. Categorization → Label instances with difficulty-level to support subsequent Refinement*.

**Categorization** refers to annotating difficulty levels for all instances, which is essential for subsequent targeted **Refinement**. To implement this, all instances are independently answered by multiple top-tier LLMs. Based on their aggregated performance, instances are classified into three difficulty tiers: easy, medium, and hard according to predefined thresholds. For outlier cases such as inconsistent model performances or ambiguous instances, experts perform coarse-grained quality control by manual difficulty annotation and calibration. This stage yields **a difficulty-stratified instance pool**, establishing the essential foundation for subsequent **Data Refinement**.

## 3.2 DATA REFINEMENT

For improving the data quality of EESE-Pool, we establish **Refinement**, which minimizes easy/medium- instances while amplifying high-difficulty ones.

This stage begins with a systematic check, which identifies instances requiring revision (primarily targeting easy-level instances, but also covering medium and high-difficulty ones). Instances marked for revision undergo additional analysis of the proportion of key information, the extent of cross-disciplinary knowledge, and the cognitive dimensions. Based on the analysis results, they are routed into a **Parallel Three-Branch Refinement Framework**: *Enhancement By Distraction*, *Enrichment By Cross-Disciplinary*, and *Refinement By Expert-Driven*, depending on the level of **Human Involvement (HI)** shown in Figure 3.

**Enhancement By Distraction (Low HI)** increases instance difficulty by introducing plausible yet misleading information to test the attention and discrimination abilities of model (Qu et al., 2024; Zhang et al., 2024; Wang et al., 2025b). This approach facilitates the

transformation of simple instances into more robust measures of fine-grained reasoning (Çavuşoğlu et al., 2024; Parikh et al., 2025). In application, multiple-choice instances receive high-quality distractors that appear credible but are incorrect, while open-ended instances include extraneous details that must be filtered out. Most distractors are auto-generated and undergo experts verify correctness and relevance (fine-grained quality control). Overall, this method efficiently elevates question difficulty with low HI.

**Enrichment By Cross-Disciplinary (Medium HI)** incorporates contexts or concepts from other field to add difficulty. This strategy is effective since tasks requiring knowledge integration across fields impose greater cognitive demands than single-field tasks (Skulmowski & Xu, 2022; Chen et al., 2024; Knar, 2025; Zhou et al., 2025c; Guo et al., 2025). Typically, initial interdisciplinary content is generated by LLMs, followed by a fine-grained review and refinement by experts to ensure factual precision and educational alignment. This method raises instance difficulty through multi-field scenarios with medium HI.

**Expert-Driven Refinement (High HI)** entails manual rewriting or restructuring by human experts to enhance clarity, embed subtle complexity, or decompose multi-step reasoning. This process is essential for instances that require nuanced logical relationships or interdisciplinary synthesis. All revisions are performed manually and undergo fine-grained quality validation to ensure consistency with targeted difficulty and scientific rigor. In summary, this method guarantees instance quality through high HI.

In summary, the **Refinement** systematically increases instance difficulty through the Parallel Three-Branch Refinement Framework, transforming candidate instances into a more scientific EESE-Pool.

### 3.3 EESE From EESE-Pool

To tackle the issues of leakage risk and evaluation inefficiency, we design EESE as a dynamic benchmark derived from the large-scale EESE-Pool. Specifically, we periodically resample 500 instances from the EESE-Pool to create a new EESE, ensuring its continued representativeness. By periodically sampling and strictly verifying, EESE ensures that each release remains fresh, robust, and difficult to leakage into training data. Unlike static benchmarks, this evolving mechanism makes EESE far more resilient against data leakage and evaluation inefficiency.

Meanwhile, although EESE inherits the core principles of Range, Reach, and Rigor from the EESE-Pool, these design factors are intentionally balanced to serve the primary goal: providing a trustworthy, low-cost, and leakage-resistant scientific benchmark that better reflects real-world model generalization.

## 4 EXPERIMENT RESULTS

### 4.1 BENCHMARK CANDIDATES

To ensure the results are comprehensive and up-to-date, we select 32 competitive LLMs for evaluation, including open-source, proprietary closed-source, and thinking-series models. Specifically, the leading proprietary models includes O3 (OpenAI, 2025b), O3-mini (OpenAI, 2025b), GPT-4o (OpenAI, 2024), GPT-4.1 (OpenAI, 2025a) from OpenAI, Gemini-2.5-pro (Gemini Team, Google DeepMind, 2025) and Gemini-1.5-pro (Team et al., 2024) from Google, Claude-3-5-sonnet (Anthropic, 2024) from Anthropic, Grok-4 (xAI Team, 2025b), Grok-2 (xAI Team, 2024) and Grok-3 (xAI Team, 2025a), as well as other popular models (Bai et al., 2023; Mistral AI Team, 2024). The open-source models cover DeepSeek-R1 (DeepSeek-AI and collaborators, 2025), Qwen3-235b-A22b (Yang et al., 2025), Qwen2.5-72B-Instruct (Yang et al., 2024), Qwen2.5-32B (Yang et al., 2024), GLM-4-32B (GLM et al., 2024), InternLM Cai et al. (2024); Team (2025), Llama-3 series (Grattafiori et al., 2024), Gemma-3 (Team et al., 2025), and Phi-4-mini (Microsoft et al., 2025). Thinking-series models such as O3, Grok-4, and Gemini-2.5-pro serve as optimized reference points for evaluating the trade-off between performance and deployment costs. All LMMs are tested with

Table 1: Performance comparison of human experts and 32 open- and closed-source LLMs on EESE across five disciplines and overall scores. Top three performance are highlighted (Best in **bold**, second and third best underlined). 'Org.' denotes the organization. 'Params.' is the parameter number. 'Open.' indicates open-sourced situation.

| Model | Model Attribute | | | Evaluation Dimensions | | | | | |
|---|---|---|---|---|---|---|---|---|---|
| | *Org.* | *Params* | *Open.* | *SSH* | *AS* | *MS* | *NS* | *ETS* | *Overall* |
| *Human* | | | | | | | | | |
| Expert | / | / | / | 0.9030 | 0.7950 | 0.8310 | 0.8815 | 0.8260 | 0.8473 |
| *Models With Thinking* | | | | | | | | | |
| O3 | *OpenAI* | *N/A* | ✗ | 0.3686 | 0.5121 | 0.4041 | **0.3922** | 0.3865 | **0.4025** |
| Gemini-2.5-pro | *Google* | *N/A* | ✗ | 0.2629 | **0.5414** | **0.4276** | 0.3640 | **0.3892** | 0.3813 |
| Grok-4 | *xAI* | *N/A* | ✗ | **0.3829** | 0.3431 | 0.3357 | 0.3160 | 0.3480 | 0.3442 |
| Deepseek-R1 | *Deepseek* | *671B* | ✔ | 0.2600 | 0.3431 | 0.3428 | 0.3632 | 0.3180 | 0.3251 |
| O3-mini | *OpenAI* | *N/A* | ✗ | 0.2438 | 0.4034 | 0.2327 | 0.3848 | 0.2926 | 0.3068 |
| Qwen3-235B-A22B | *Alibaba Cloud* | *235B* | ✔ | 0.2105 | 0.2397 | 0.2510 | 0.2848 | 0.2740 | 0.2543 |
| *Models Without Thinking* | | | | | | | | | |
| Claude-3-7-sonnet | *Anthropic* | *N/A* | ✗ | 0.2486 | 0.2655 | 0.2429 | 0.2304 | **0.3461** | **0.2648** |
| Deepseek-V3 | *DeepSeek* | *671B* | ✔ | 0.2019 | 0.2431 | 0.2551 | **0.2624** | 0.3197 | 0.2572 |
| Claude-3-5-sonnet | *Anthropic* | *N/A* | ✗ | 0.2591 | 0.1948 | 0.2633 | 0.2049 | 0.3274 | 0.2521 |
| GPT-4.1 | *OpenAI* | *N/A* | ✗ | 0.2419 | 0.3603 | 0.2837 | 0.2112 | 0.2176 | 0.2514 |
| GPT-4o | *OpenAI* | *N/A* | ✗ | 0.2029 | 0.2448 | 0.3041 | 0.2216 | 0.2354 | 0.2397 |
| Grok-2 | *xAI* | *N/A* | ✗ | 0.2771 | 0.2224 | 0.1796 | 0.2184 | 0.2841 | 0.2372 |
| Qwen2.5-VL-32B-Instruct | *Alibaba Cloud* | *32B* | ✔ | 0.2194 | 0.2345 | 0.2286 | 0.1736 | 0.2540 | 0.2183 |
| Qwen-vl-max | *Alibaba Cloud* | *N/A* | ✗ | 0.2114 | 0.2448 | 0.2041 | 0.1784 | 0.2540 | 0.2142 |
| Gemini-1.5-pro | *Google* | *N/A* | ✗ | 0.2401 | 0.2793 | 0.1173 | 0.2040 | 0.2334 | 0.2093 |
| GLM-4-32B | *Zhipu AI* | *4B* | ✔ | 0.2194 | 0.2052 | 0.2347 | 0.1623 | 0.2202 | 0.2056 |
| Qwen2.5-32B-Instruct | *Alibaba Cloud* | *32B* | ✔ | 0.2114 | 0.2724 | 0.1898 | 0.1288 | 0.2548 | 0.2019 |
| Grok-3 | *xAI* | *N/A* | ✗ | 0.2210 | 0.1759 | 0.1735 | 0.1752 | 0.2493 | 0.1998 |
| Mistral-large | *Mistral AI* | *N/A* | ✗ | 0.2011 | 0.2069 | 0.1694 | 0.1768 | 0.2368 | 0.1963 |
| Qwen2.5-72B-Instruct | *Alibaba Cloud* | *72B* | ✔ | 0.1914 | 0.2466 | 0.1694 | 0.1617 | 0.2410 | 0.1957 |
| Qwen2.5-VL-72B-Instruct | *Alibaba Cloud* | *72B* | ✔ | 0.2057 | 0.2172 | 0.1694 | 0.1456 | 0.2610 | 0.1955 |
| Phi-4 | *Microsoft* | *14B* | ✔ | 0.1829 | 0.2052 | 0.2012 | 0.1304 | 0.2134 | 0.1817 |
| Internlm3-8b-instruct | *OpenGVLab* | *8B* | ✔ | 0.1438 | 0.2034 | 0.2031 | 0.1123 | 0.2441 | 0.1745 |
| Llama-3.3-70B-Instruct | *Meta* | *70B* | ✔ | 0.1819 | 0.1776 | 0.1408 | 0.1504 | 0.2024 | 0.1691 |
| Llama-3.1-70B-Instruct | *Meta* | *70B* | ✔ | 0.1724 | 0.2345 | 0.1490 | 0.1216 | 0.1691 | 0.1613 |
| gemma-3-27b-it | *Gemma Team* | *27B* | ✔ | 0.1914 | 0.1569 | 0.1327 | 0.1448 | 0.1432 | 0.1535 |
| Internlm2.5-20b-chat | *OpenGVLab* | *20B* | ✔ | 0.1486 | 0.1724 | 0.1388 | 0.1256 | 0.1833 | 0.1545 |
| internlm2-chat-20b | *OpenGVLab* | *20B* | ✔ | 0.1219 | 0.1672 | 0.0982 | 0.0984 | 0.1603 | 0.1243 |
| Llama-3.2-11B-Vision-Instruct | *Meta* | *11B* | ✔ | 0.1524 | 0.0862 | 0.1122 | 0.0847 | 0.1443 | 0.1152 |
| Llama-3.1-8B-Instruct | *Meta* | *8B* | ✔ | 0.1314 | 0.1172 | 0.1092 | 0.1024 | 0.0887 | 0.1088 |
| Internlm2.5-7b-chat | *OpenGVLab* | *7B* | ✔ | 0.1695 | 0.1001 | 0.1306 | 0.0648 | 0.0675 | 0.1053 |
| Phi-4-mini-instruct | *Microsoft* | *3.8B* | ✔ | 0.1429 | 0.0828 | 0.0469 | 0.0824 | 0.0881 | 0.0895 |

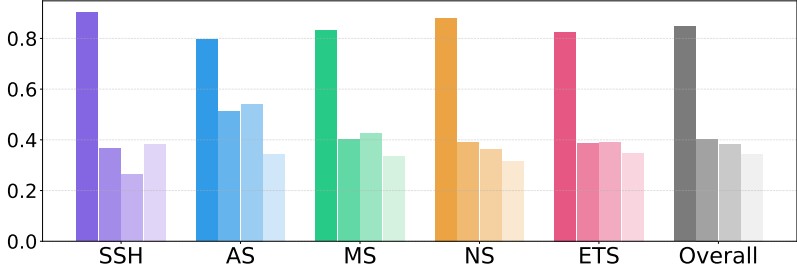

Figure 5: Quick comparison of human performance and top-performing *models with thinking* on EESE. Each bar group corresponding to the specific discipline represents the scores of Human, O3, Gemini-2.5-Pro, and Grok-4 (from left to right) respectively.

zero-shot setting. In addition, the average accuracy of 10 experts is recorded to illustrate performance differences.

## 4.2 Performance Analysis

**I. EESE-Pool demonstrates significant disciplinary variations across models while exposing their limitations in scientific abilities**. Figure 4 presents the performance distribution of six representative models on EESE-Pool. The results reveal significant discipline-specific variations. Crucially, no single model establishes comprehensive superiority across all disciplines. Besides, the average accuracy of the six models remains low, highlighting the challenges of scientific questions for current foundation models. Overall, the results confirm that EESE-Pool effectively reveals nuanced weaknesses in scientific questions, and serves as comprehensive question pool for robustly differentiating model capabilities.

Table 2: Speed, cost, and performance comparison on EESE between top *models with thinking* and the best *models without thinking* from Anthropic, DeepSeek, and OpenAI. '×' denotes relative value to the (best models without thinking). Speed: avg. inference time/question (s). Cost: avg. cost per 10 questions (USD).

| Model | Model Attribute | | | Evaluation Dimensions | | |
|---|---|---|---|---|---|---|
| | *Org.* | *Params* | *Open.* | *Speed$_{s/q}$* | *Cost$_{\$/10q}$* | *Overall (EESE)* |
| *Models With Thinking* | | | | | | |
| O3 | *OpenAI* | *N/A* | ✗ | $15.100_{\times 1.064}$ | $0.125_{\times 2.551}$ | $0.4025_{\times 1.561}$ |
| Gemini-2.5-pro | *Google* | *N/A* | ✗ | $19.570_{\times 1.379}$ | $0.442_{\times 9.001}$ | $0.3813_{\times 1.479}$ |
| Grok-4 | *xAI* | *N/A* | ✗ | $41.450_{\times 2.920}$ | $0.440_{\times 8.943}$ | $0.3442_{\times 1.335}$ |
| Deepseek-R1 | Deepseek | 671B | ✔ | $107.480_{\times 7.572}$ | $0.039_{\times 0.786}$ | $0.3251_{\times 1.261}$ |
| O3-mini | *OpenAI* | *N/A* | ✗ | $7.240_{\times 0.510}$ | $0.048_{\times 0.972}$ | $0.3068_{\times 1.190}$ |
| Qwen3-235B-A22B | *Alibaba Cloud* | 235B | ✔ | $79.000_{\times 5.566}$ | $0.058_{\times 1.178}$ | $0.2543_{\times 0.986}$ |
| Average | / | / | / | $44.973_{\times 4.243}$ | $0.192_{\times 4.492}$ | $0.3357_{\times 1.302}$ |
| *Models Without Thinking* | | | | | | |
| Claude-3-7-sonnet | *Anthropic* | *N/A* | ✗ | $10.400_{\times 0.733}$ | $0.106_{\times 2.155}$ | $0.2648_{\times 1.027}$ |
| Deepseek-V3 | *DeepSeek* | 671B | ✔ | $24.000_{\times 1.691}$ | $0.006_{\times 0.116}$ | $0.2572_{\times 0.998}$ |
| GPT-4.1 | *OpenAI* | *N/A* | ✗ | $9.082_{\times 0.640}$ | $0.036_{\times 0.729}$ | $0.2514_{\times 0.975}$ |
| Average | / | / | / | $14.194_{\times 1.000}$ | $0.0491_{\times 1.000}$ | $0.2578_{\times 1.000}$ |

**II. EESE reveals that models with thinking and proprietary designs tend to perform better, yet clear discipline-specific weaknesses, substantial gaps between models and humans, and the high quality of EESE remain evident**. Table 1 and Figure 5 provide a quantitative comparison and a quick visualized comparison between human experts and 32 leading Large Language Models (LLMs), covering 5 disciplines.

From the results, several findings can be drawn. First, models with thinking consistently outperform models without thinking, demonstrating the benefit of thinking-augmented design. Second, closed-source models generally score higher than open-source ones, likely due to proprietary data, tuning strategies, or infrastructure. Third, large discipline-specific gaps persist, as no model excels uniformly across all scientific fields, highlighting ongoing challenges in specialized or interdisciplinary areas. Fourth, a considerable performance gap persists between even the best models and human experts. Overall, the clear and consistent performance differences confirm that EESE is sufficiently challenging and discriminative to reveal meaningful gaps in scientific proficiency.

**III. Though models with thinking achieve better performance, their overall cost-effectiveness remains limited**. Table 2 provides a comparative overview of inference efficiency (Speed), economic cost (Cost), and performance (Overall) between models with thinking and the best models without thinking. To better highlight the advantages and limitations, we use the average of the best models without thinking as baseline.

Table 2 highlights several key observations. First, models with thinking consistently outperform models without thinking, which confirms that thinking possibly improving instance difficulty. Second, the efficiency trade-offs are significant. Models with thinking take about 4.2× longer and 4.5× more, only improve performance by 1.3× compared to models without thinking. This imbalance suggests that the marginal gains may not justify the extra cost and burden, raising concerns about the practicality of high-difficulty approaches in real-world deployments. Third, even the best-performing models with thinking far below human expert performance. This further illustrates the high quality and substantial difficulty of the EESE.

**IV. EESE serves as a representative, low-cost proxy for the EESE-Pool**. Figure 6 (a) presents the spearman rank-order correlation coefficient (SRCC) (Wang et al., 2025a; Zhang et al., 2025a) heatmap within EESE, covering five disciplines and the overall score. Figure 6 (b) displays the SRCC heatmap between EESE and EESE-Pool across the same disciplines. The SRCC is calculated by ranking models based on the performance in each discipline and then computing the Spearman correlation between these rankings.

As shown in Figure 6 (a), the consistently high SRCC values indicate strong internal consistency and balanced instance coverage. As shown in Figure 6 (b), the high diagonal values indicates that the rankings derived from the EESE closely match those from the 100K+

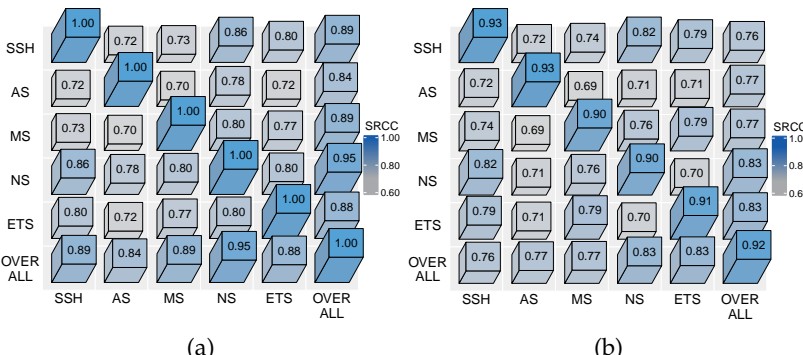

(a)  (b)

Figure 6: Discipline correlation heatmaps with spearman rank-order correlation coefficient (SRCC). (a) shows internal correlations of EESE across five disciplines and overall scores (*X-axis → EESE, Y-axis → EESE*) while (b) presents the discipline correlations between EESE (*X-axis*) and EESE-Pool (*Y-axis*).

Table 3: Comparison of model performance *Before* and *After* **Refinement** on EESE.

| Model | *Before* **Refinement** | | | *After* **Refinement** | | |
|---|---|---|---|---|---|---|
| | *AS* | *MS* | *overall* | *AS* | *MS* | *overall* |
| O3 | **0.6214** | 0.5134 | **0.5218** | 0.5121 | 0.4041 | **0.4025** |
| Gemini-2.5-pro | 0.6201 | **0.5243** | 0.4880 | **0.5414** | **0.4276** | 0.3813 |
| Deepseek-R1 | 0.4545 | 0.3398 | 0.4332 | 0.3431 | 0.3428 | 0.3251 |
| O3-mini | 0.5001 | 0.3294 | 0.4035 | 0.4034 | 0.2327 | 0.3068 |
| Claude-3-7-sonnet | 0.3622 | 0.3396 | 0.3615 | 0.2655 | 0.2429 | 0.2648 |
| Deepseek-V3 | 0.3398 | 0.3518 | 0.3539 | 0.2431 | 0.2551 | 0.2572 |
| Claude-3-5-sonnet | 0.2915 | 0.3600 | 0.3488 | 0.1948 | 0.2633 | 0.2521 |

EESE-Pool for each corresponding discipline, confirming that EESE reliably reflects the performance trends of broader benchmark. In summary, EESE is a reliable, low-cost and leak-resistant proxy for EESE-Pool, which faithfully reflects the EESE-Pool's ability to differentiate the science capabilities of models.

**V. Refinement successfully increases the instance quality**. As shown in Table 3, all representative models exhibit lower accuracy after refinement across disciplines and the overall score. This consistent decrease confirms that the refinement effectively increases instance difficulty and reduces trivial or overly simple items. By additional plausible distractors, interdisciplinary contexts, and expert-driven rewrite, the refined EESE instances impose higher quality. This leads to clearer performance gaps among models, and demonstrates that EESE achieves the intended rigor while maintaining reliability for evaluation.

## 5 CONCLUSION

In this work, we present EESE, a dynamic benchmark that systematically balances Range, Reach, and Rigor through a large, high-quality EESE-Pool (constructed via a multi-stage Data Engine and a three-branch Data Refinement process). By periodically sampling and updating, EESE minimize leakage risks and evaluation inefficiency while remaining representative of the larger pool. Extensive experiments show that EESE effectively raises instance difficulty, exposes significant performance differences across disciplines, and highlights trade-offs between inference cost and science ability. In addition, we show that benchmark developers no longer need to choose between scale and security: the two-level EESE design provides a practical way to continually refresh test sets, adapt to evolving model capabilities, and sustain benchmark difficulty over time. More broadly, EESE demonstrates how a dynamic, well-curated benchmark can reveal subtle differences in science evaluation, drive the development of more robust models, and serve as a practical blueprint for building more trustworthy benchmarks.

## 6 ETHICS STATEMENT

This work adheres to the ICLR Code of Ethics. In this study, no human subjects or animal experimentation was involved. All datasets used, including EESE, are sourced in compliance with relevant usage guidelines, ensuring no violation of privacy. We have taken care to avoid any biases or discriminatory outcomes in our research process. No personally identifiable information is used, and no experiments are conducted that could raise privacy or security concerns. We are committed to maintaining transparency and integrity throughout the research process.

## 7 REPRODUCIBILITY STATEMENT

We have made every effort to ensure that the results presented in this paper are reproducible. We guarantee that all relevant code and datasets will be made publicly available, thereby enabling the research community to replicate and verify our findings. The experimental setup, including training steps, model configurations, and hardware details, is described in detail in the paper. We have also provided a full description of EESE to assist others in reproducing our experiments.

Additionally, datasets used in the paper are publicly available, ensuring consistent and reproducible evaluation results.

We believe these measures will enable other researchers to reproduce our work and further advance the field.

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

## A    LLM Usage

Large Language Models (LLMs) are used to aid in the writing and polishing of the manuscript. Specifically, we use an LLM to assist in refining the language, improving readability, and ensuring clarity in various sections of the paper. The model helps with tasks such as sentence rephrasing, grammar checking, and enhancing the overall flow of the text.

It is important to note that the LLM is not involved in the ideation, research methodology, or experimental design. All research concepts, ideas, and analyses are developed and conducted by the authors. The contributions of the LLM are solely focused on improving the linguistic quality of the paper, with no involvement in the scientific content or data analysis.

The authors take full responsibility for the content of the manuscript, including any text generated or polished by the LLM. We ensure that the LLM-generated text adheres to ethical guidelines and does not contribute to plagiarism or scientific misconduct.

## B    Refinement Methods and Optimization Analysis

The refinement methods can be categorized into three levels: (1) Enhancement by Distraction (Low HI), (2) Enrichment by Cross-Disciplinary Integration (Medium HI), and (3) Expert-Driven Refinement (High HI). Below, we present examples of each method, comparing the original questions with their refined versions and analysing the enhancement process. The differences between the original and refined versions are highlighted in red.

1. Enhancement by Distraction (Low HI)

---

**Original Question**

**Original Question:** A protocol suite is ()=.
- A) A set of protocols.
- B) A hierarchical collection of protocols.

---

**Refined Question**

**Refined Question:** Regarding protocol suites, which of the following statements is correct? ()
- A) TCP/IP only includes two protocols.
- B) The OSI model has seven layers, each corresponding to a specific protocol in a suite.
- C) HTTP is not part of any protocol suite.
- D) FTP operates independently of any protocol suite.

---

**Analysis:** The refined version builds upon the original options by expanding the abstract terms "a set of protocols" and "a hierarchical collection of protocols" through the inclusion of concrete examples such as TCP/IP, OSI, HTTP, and FTP. This enrichment enhances the informational depth of the question and elevates its professional and technical specificity, thereby reducing the probability of correct guessing and strengthening the assessment's effectiveness in evaluating true conceptual understanding.

2. ENRICHMENT BY CROSS-DISCIPLINARY INTEGRATION (MEDIUM HI)

---

**Original Question**

**Original Question:** Given an element with a maximum oxidation state of +7, determine its period and group.

---

**Refined Question**

**Refined Question:** Elements A, B, C, and D are from period 4:

- A forms a 1:1 compound with an oxidation state of +7;
- B is a d-block element with an oxidation state of +7;
- C is in the same period and has the same oxidation state as B;
- D is the most electronegative element in the set.

Fill in the table below and order the four elements by electronegativity from high to low.

| | Element | Symbol | Period | Group | Max Oxidation |
|---|---|---|---|---|---|
| A | | | | | |
| B | | | | | |
| C | | | | | |
| D | | | | | |

Table 4: Element Properties

---

**Analysis:** The refined question improves upon the original by integrating cross-disciplinary knowledge and contextual clues, promoting higher-order thinking. While the original question only asked students to identify the period and group of an element with a +7 oxidation state—requiring basic recall—the revised version introduces four elements from period 4, each with specific properties tied to oxidation states, electronegativity, and element classification. Students must analyze multiple clues, apply periodic trends, and reconcile inconsistencies (e.g., fluorine not being in period 4), which fosters critical thinking. They also complete a table and rank elements by electronegativity, combining factual knowledge with synthesis and evaluation. This enhancement increases cognitive demand, integrates multiple chemistry concepts, and reduces guessing, transforming a simple recall question into a comprehensive reasoning task.

3. EXPERT-DRIVEN REFINEMENT (HIGH HI)

---

**Original Question**

**Original Question:** A machine has a 16-bit instruction field and a 6-bit address field. If the opcode is 8 bits long, how many 0-address instructions are possible?

---

**Refined Question**

**Refined Question:** A machine uses 16-bit instruction words and 6-bit operand addresses. Assume the opcode length is fixed, with instructions in three formats: 0-, 1-, and 2-address. Given M 0-address and N 1-address instructions, what is the maximum number of 2-address instructions? If the opcode length is variable, what is the maximum number of 2-address instructions?

---

**Analysis:** The refined question improves upon the original by introducing multiple instruction formats (0-, 1-, and 2-address) and asking students to calculate the maximum number of 2-address instructions under both fixed and variable opcode length assumptions. This requires a deeper understanding of instruction encoding and opcode space management. Unlike the original, which involved a simple calculation based on fixed field sizes, the enhanced version tests students' ability to analyze how opcode and address fields are shared across different instruction types, apply multi-step reasoning to maximize opcode space under architectural constraints, and understand advanced encoding techniques such as opcode expansion in variable-length models. By embedding theoretical concepts into a practical design problem, the question promotes higher-order thinking and better assesses students' grasp of computer architecture principles.

## C  DIFFICULTY-STRATIFIED SAMPLES

### 💡 Easy Sample

**Question:**
Regarding the structures of PROM and PAL, which of the following statements are correct? ()

    A) PROM has a fixed AND array that is not programmable

    B) Both AND array and OR array of PROM are not programmable

    C) Both AND array and OR array of PAL are programmable

    D) The AND array of PAL is programmable

**Answer:** AD
**Discipline:** Engineering and Technological Sciences
**Field:** Electronics and Communication Technology
**Subfield:** Electronic Technology

**Question:**
According to the causes of dyspnea and its manifestations, dyspnea can be divided into _________________, _________________, _________________ three types.
**Answer:** inspiratory dyspnea, expiratory dyspnea, mixed dyspnea
**Discipline:** Agricultural Sciences
**Field:** Animal Husbandry and Veterinary Science
**Subfield:** Veterinary Medicine

**Question:**
The main issues to note when designing a social survey research plan are ( ).
A. Practicality   B. Systematicness   C. Timeliness   D. Economy   E. Accuracy   F. Flexibility
**Answer:** ABCDF
**Discipline:** Social Sciences and Humanities
**Field:** Sociology
**Subfield:** Sociological Methods

**Question:**
Among the following drugs, those with optical activity are ( )
A. Ranitidine   B. Ephedrine   C. Pethidine   D. Omeprazole   E. Naproxen
**Answer:** ABCDE
**Discipline:** Medical Sciences
**Field:** Pharmacy
**Subfield:** Medicinal Chemistry

**Question:**
Judge whether the following statement is correct: According to the change law of the resistance coefficient along the path, the Nikuradse experimental curve is divided into three regions.
**Answer:** False
**Discipline:** Natural Sciences
**Field:** Mechanics
**Subfield:** Fluid Mechanics

### 🔍 Middle Sample

**Question:** The Foreign Trade Import and Export Service Company under the Foreign Trade Bureau of City A signed a sales contract with Enterprise B of City A. A dispute arose during the performance of the contract. Later, the Foreign Trade Import and Export Service Company was divided into two separate legal entities: the Foreign Trade Commodity Trading Company of City A and the Import and Export Service Company of City A. No arrangements were made regarding the aforementioned sales contract during the division. Now, Enterprise B has filed a lawsuit in court over the contract dispute. The defendant(s) in this lawsuit should be ( )

    A) The Foreign Trade Import and Export Service Company of City A

    B) The Foreign Trade Bureau of City A

    C) Either the Foreign Trade Commodity Trading Company of City A or the Import and Export Service Company of City A

    D) Both the Foreign Trade Commodity Trading Company of City A and the Import and Export Service Company of City A

**Answer:** C
**Discipline:** Social Sciences and Humanities
**Field:** Law
**Subfield:** Sectoral Law

**Question:**
Determine whether the following statement is correct: Both the in-duct dilution probe and the out-of-duct dilution probe use critical sonic orifice sampling.
**Answer:** False
**Discipline:** Engineering and Technological Sciences

**Field:** Environmental Science and Technology and Resource Science and Technology
**Subfield:** Environmental Engineering

**Question:**
The damage caused by above-zero low temperature to thermophilic plants is generally divided into two steps:
Step 1: ______________________, Step 2: ______________________
**Answer:** Change in membrane phase / Membrane phase transition; Death resulting from metabolic disorder due to membrane damage
**Discipline:** Agricultural Sciences
**Field:** Agronomy
**Subfield:** Basic Agricultural Sciences

**Question:**
What is the natural reaction method? What is its application value in infant research?
**Answer:**
1. Definition: By examining the innate reflex activities of infants and young children, make inferences on the development and changes of their psychological abilities and their essence.
2. Application value:
- Many innate reflexes have important survival value
- Typical examples: visual tracking and cliff response
**Discipline:** Natural Sciences
**Field:** Psychology
**Subfield:** Developmental Psychology

**Question:**
Which of the following statements about weighted imaging is correct?

    A) T1WI is the T1 value map of tissue

    B) Proton density affects signal intensity in any pulse sequence image

    C) The longer the T1 value of tissue, the higher the signal on T1WI

    D) The longer the T2 value of tissue, the lower the signal intensity

    E) T2WI refers to imaging parameters that extend the tissue's T2 value

**Answer:** A
**Discipline:** Medical Sciences
**Field:** Basic Medical Sciences
**Subfield:** Radiology

**💣 Hard Sample**

**Question:**

A certain machine has an instruction word length of 16 bits, and each operand's address code is 6 bits. Assume the opcode length is fixed, and instructions are divided into three formats: zero-address, one-address, and two-address. If there are M zero-address instructions and N one-address instructions, what is the maximum number of two-address instructions? If the opcode length is variable, what is the maximum number of two-address instructions allowed?

**Answer:**

1) If a fixed-length opcode is used, the two-address instruction format is as follows: Let $K$ be the number of two-address instructions. Then

$$K = 2^4 - M - N$$

When $M = 1$ (minimum) and $N = 1$ (minimum), the maximum number of two-address instructions is

$$K_{\max} = 16 - 1 - 1 = 14.$$

2) If a variable-length opcode is used, the two-address instruction format is still as shown in 1), but the opcode length can vary with the number of address codes. In this case,

$$K = 2^4 - \left( \frac{N}{2^6} + \frac{M}{2^{12}} \right).$$

When $\frac{N}{2^6} + \frac{M}{2^{12}} \leq 1$, $K$ is maximized. So the maximum number of two-address instructions is

$$K_{\max} = 16 - 1 = 15$$

(leaving one encoding as an extension flag).

**Discipline:** Engineering and Technological Sciences
**Field:** Computer Science and Technology
**Subfield:** Computer System Architecture

**Question:**

It is known that two of the following four statements are true.
1) Everyone in Class A is from Shanghai.
2) Zhao Yun in Class A is from Shanghai.
3) Some people in Class A are from Shanghai.
4) Some people in Class A are not from Shanghai.
Question: Can we determine whether Zhao Yun in Class A is from Shanghai?

**Answer:** Cannot be determined
**Discipline:** Social Sciences and Humanities
**Field:** Philosophy
**Subfield:** Logic

**Question:**

The pharmacological effects of thiazide diuretics include: ( )

    A) Antihypertensive effect

    B) Decrease in glomerular filtration rate

    C) Increase in blood glucose levels

    D) Increase in urate excretion

    E) Antidiuretic effect

**Answer:** ABCE

**Discipline:** Agricultural Sciences

**Field:** Animal Husbandry and Veterinary Science

**Subfield:** Veterinary Medicine

---

**Question:**

Some Nocardia species are acid-fast positive, but only with _________________.

Prolonged decolorization renders them negative, which helps differentiate them from _________________ bacteria. **Answer:** 1% hydrochloric acid ethanol; Mycobacterium tuberculosis

**Discipline:** Medical Sciences

**Field:** Basic Medical Sciences

**Subfield:** Medical Microbiology

---

**Question:**

Suppose $\{N(t), t \geq 0\}$ is a Poisson process with intensity $\lambda$, $X_n$ $(n \geq 1)$ represents the time interval between the $(n-1)$st and $n$th event, then $\mathbb{E}(X_1 \mid N(t) = 1) = $ __________.

**Answer:** $t/2$

**Discipline:** Natural Sciences

**Field:** Mathematics

**Subfield:** Probability Theory

# D  THE SUBFIELD OF EESE-POOL

## 1. NATURAL SCIENCES

| Natural Sciences | |
|---|---|
| **Field** | **Subfield** |
| | A1: History of Mathematics (35) |
| | A2: Algebra (48) |
| | A3: Geometry (34) |
| | A4: Function Theory (155) |
| | A5: Ordinary Differential Equations (207) |
| | A6: Probability Theory (263) |
| | A7: Mathematical Statistics (80) |
| | A8: Discrete Mathematics (79) |
| | A9: Mathematical Logic and Foundations (80) |
| Mathematics | A10: Number Theory (80) |
| | A11: Algebraic Geometry (80) |
| | A12: Topology (80) |
| | A13: Mathematical Analysis (85) |
| | A14: Integral Equations (81) |
| | A15: Applied Statistical Mathematics (80) |
| | A16: Operations Research (80) |
| | A17: Combinatorial Mathematics (80) |
| | A18: Fuzzy Mathematics (80) |
| | A19: Computational Mathematics (80) |
| | A20: Applied Mathematics (80) |
| | A21: Basic Disciplines of Information Science and Systems Science (120) |
| Information Science and Systems Science | A22: Systems Science (73) |
| | A23: Control Theory (80) |
| | A24: System Evaluation and Feasibility Analysis (80) |
| | A25: Systems Engineering Methodology (72) |
| Mechanics | A26: Basic Mechanics (141) |
| | A27: Fluid Mechanics (1334) |
| | A28: History of Physics (23) |
| | A29: Theoretical Physics (59) |
| | A30: Acoustics (25) |
| | A31: Thermodynamics (488) |
| | A32: Optics (30) |
| Physics | A33: Electromagnetism (404) |
| | A34: Electronic Physics (108) |
| | A35: Condensed Matter Physics (95) |
| | A36: Atomic and Molecular Physics (85) |
| | A37: Computational Physics (35) |
| | A38: Applied Physics (202) |
| | A39: Inorganic Chemistry (156) |
| | A40: Organic Chemistry (24) |
| | A41: Analytical Chemistry (31) |
| | A42: Physical Chemistry (604) |
| | A43: Polymer Physics (30) |
| | A44: Materials Chemistry (61) |
| Chemistry | A45: History of Chemistry (86) |
| | A46: Chemical Physics (70) |
| | A47: Polymer Chemistry (71) |
| | A48: Nuclear Chemistry (80) |
| | A49: Applied Chemistry (80) |

| Natural Sciences | |
|---|---|
| **Field** | **Subfield** |
| | A50: Celestial Mechanics (72) |
| | A51: Astrophysics (70) |
| | A52: Cosmochemistry (70) |
| | A55: Galaxies and Cosmology (80) |
| Astronomy | A53: Stellar Evolution (80) |
| | A54: Stars and the Milky Way (80) |
| | A56: The Sun and Solar System (76) |
| | A57: Astrogeodynamics (80) |
| | A58: Chronometry (80) |
| | A59: Geology (153) |
| | A60: Atmospheric Science (70) |
| | A61: Solid Earth Geophysics (80) |
| | A62: Space Physics (80) |
| | A63: Geochemistry (80) |
| Earth Science | A64: Geodesy (80) |
| | A65: Cartography (79) |
| | A66: Geography (80) |
| | A67: Hydrology (77) |
| | A68: Ocean Science (82) |
| | A69: Biophysics (21) |
| | A70: Biochemistry (48) |
| | A71: Cell Biology (70) |
| | A72: Immunology (42) |
| | A73: Physiology (108) |
| | A74: Developmental Biology (171) |
| | A75: Genetics (43) |
| | A76: Molecular Biology (67) |
| | A77: Evolutionary Biology (44) |
| Biology | A78: Ecology (565) |
| | A79: Neurobiology (46) |
| | A80: Botany (1697) |
| | A81: Entomology (734) |
| | A82: Zoology (1007) |
| | A83: Microbiology (513) |
| | A84: Virology (22) |
| | A85: Anthropology (21) |
| | A86: Social Psychology (167) |
| | A87: Developmental Psychology (916) |
| | A88: Psychometrics (366) |
| | A89: Physiological Psychology (454) |
| Psychology | A90: Managerial Psychology (169) |
| | A91: Educational Psychology (319) |

2. AGRICULTURAL SCIENCE

| Agricultural Science | |
|---|---|
| **Field** | **Subfield** |
| | B1: Basic Agricultural Sciences (1136) |
| | B2: Crop Science (84) |
| | B3: History of Agriculture (90) |
| Agronomy | B4: Horticulture (79) |
| | B5: Storage and Processing of Agricultural Products (75) |
| | B6: Soil Science (76) |
| | B7: Plant Protection Science (79) |
| | B8: Landscape Architecture (822) |
| | B9: Forest Genetics and Breeding (80) |
| | B10: Silviculture (80) |
| Forestry | B11: Forest Management (80) |
| | B12: Forest Protection (80) |
| | B13: Wildlife Conservation and Management (80) |
| | B14: Forest Statistics (80) |
| | B15: Forestry Economics (80) |
| | B16: Veterinary Medicine (1753) |
| Animal Husbandry and Veterinary Science | B17: Basic Disciplines of Animal Husbandry and Veterinary Science (80) |
| | B18: Animal Husbandry Science (80) |
| | B19: Aquafeed Science (75) |
| | B20: Aquatic Conservation (71) |
| | B21: Fisheries Science (80) |
| Aquaculture | B22: Storage and Processing of Aquatic Products (73) |
| | B23: Aquaculture Engineering (80) |
| | B24: Aquatic Resources Science (73) |

3. MEDICAL SCIENCE

| Medical Science | |
| --- | --- |
| **Field** | **Subfield** |
| Basic Medical Sciences | C1: History of Medicine (35) |
| | C2: Human Anatomy (1358) |
| | C3: Human Physiology (108) |
| | C4: Radiology (1597) |
| | C5: Medical Parasitology (159) |
| | C6: Medical Microbiology (1147) |
| | C7: Pathology (388) |
| | C8: Medical Laboratory Animal Science (247) |
| Clinical Medicine | C9: Clinical Diagnostics (90) |
| | C10: Preventive Medicine (58) |
| | C11: Anesthesiology (183) |
| | C12: Internal Medicine (549) |
| | C13: Surgery (1263) |
| | C14: Ophthalmology (514) |
| | C15: Stomatology (2186) |
| | C16: Nuclear Medicine (188) |
| | C17: General Practice (120) |
| | C18: Nursing (520) |
| Preventive Medicine and Public Health | C19: Environmental Medicine (281) |
| | C20: Health Statistics (578) |
| | C21: Nutrition (80) |
| | C22: Toxicology (75) |
| | C23: Disinfection Science (80) |
| | C24: Epidemiology (80) |
| | C25: Vector Biology Control (80) |
| | C26: Occupational Disease (80) |
| | C27: Endemic Disease (80) |
| | C28: Social Medicine (80) |
| | C29: Health Inspection (78) |
| | C30: Food Hygiene (72) |
| | C31: Environmental Hygiene (79) |
| | C32: Eugenics (80) |
| | C33: Health Promotion and Health Education (80) |
| | C34: Health Management (80) |
| Military and Special Medicine | C35: Military Medicine (70) |
| | C36: Special Medicine (72) |
| Pharmacy | C37: Medicinal Chemistry (2041) |
| | C38: Pharmaceutics (24) |
| | C39: Pharmaceutical Administration (888) |
| Traditional Chinese Medicine and Materia Medica | C40: Traditional Chinese Medicine (3226) |
| | C41: Chinese Materia Medica (2362) |

4. Engineering and Technological Sciences

| Engineering and Technological Sciences | |
|---|---|
| **Field** | **Subfield** |
| | D1: Engineering Mechanics (50) |
| | D2: Engineering Geology (81) |
| | D3: Engineering Mathematics (76) |
| | D4: Engineering Cybernetics (80) |
| Basic Disciplines of Engineering and Technological Sciences | D5: Engineering Hydrology (80) |
| | D6: Engineering Bionics (80) |
| | D7: Engineering Psychology (80) |
| | D8: Standards Science and Technology (80) |
| | D9: Metrology (80) |
| | D10: Exploration Technology (80) |
| | D11: General Engineering Technology (80) |
| | D12: Industrial Engineering (80) |
| | D13: Control Science and Technology (98) |
| Engineering and Technology Related to Information and Systems Science | D14: Information Security Technology (761) |
| | D15: Systematic Application of Information Technology (82) |
| | D16: Simulation Science and Technology (80) |
| | D17: Engineering and Technology Related to Physics (70) |
| Engineering and Technology Related to Natural Sciences | D18: Optical Engineering (125) |
| | D19: Marine Engineering and Technology (80) |
| | D20: Bioengineering (79) |
| | D21: Agricultural Engineering (83) |
| | D22: Geodetic Surveying Technology (87) |
| Surveying and Mapping Science and Technology | D23: Photogrammetry and Remote Sensing Technology (72) |
| | D24: Cartographic Technology (89) |
| | D25: Engineering Surveying Technology (540) |
| | D26: Marine Surveying (80) |
| | D27: Basic Disciplines of Materials Science (327) |
| | D28: Surveying Instruments (80) |
| | D29: Material Surfaces and Interfaces (70) |
| | D30: Material Failure and Protection (80) |
| Materials Science | D31: Material Testing and Analysis Technology (72) |
| | D32: Material Experiments (80) |
| | D33: Material Synthesis and Processing Technology (80) |
| | D34: Metallic Materials (79) |
| | D35: Inorganic Non-Metallic Materials (72) |
| | D36: Organic Polymer Materials (77) |
| | D37: Composite Materials (74) |
| | D38: Biomaterials (75) |
| | D39: Nanomaterials (80) |

| Engineering and Technological Sciences | |
|---|---|
| **Field** | **Subfield** |
| | D40: Mining Geology (88) |
| | D41: Mine Surveying (70) |
| | D42: Mine Design (75) |
| | D43: Surface Mining Engineering (78) |
| | D44: Underground Mining Engineering (80) |
| | D45: Mining Engineering (86) |
| | D46: Mineral Processing Engineering (78) |
| | D47: Drilling Engineering (80) |
| | D48: Oil and Gas Field Development Engineering (84) |
| Mining Engineering Technology | D49: Petroleum and Natural Gas Storage and Transportation Engineering (83) |
| | D50: Mining Machinery Engineering (80) |
| | D51: Mining Electrical Engineering (80) |
| | D52: Mining Environmental Engineering (87) |
| | D53: Mine Safety (93) |
| | D54: Comprehensive Utilization of Mining Resources Engineering (84) |
| | D55: Metallurgical Physical Chemistry (72) |
| | D56: Metallurgical Thermal Engineering (80) |
| Metallurgical Engineering Technology | D57: Metallurgical Technology (70) |
| | D58: Ferrous Metallurgy (70) |
| | D59: Non-Ferrous Metallurgy (70) |
| | D60: Rolling (80) |
| | D61: Metallurgical Machinery and Automation (70) |
| | D62: Mechanical Design (1941) |
| | D63: Mechanical Manufacturing Processes and Equipment (231) |
| Mechanical Engineering | D64: Cutting Tool Technology (80) |
| | D65: Machine Tool Technology (80) |
| | D66: Fluid Transmission and Control (83) |
| | D67: Mechanical Manufacturing Automation (80) |
| | D68: Electrical Engineering (681) |
| | D69: Engineering Thermophysics (80) |
| Power and Electrical Engineering | D70: Thermal Engineering (80) |
| | D71: Power Machinery Engineering (80) |
| | D72: Refrigeration and Cryogenic Engineering (80) |

| Engineering and Technological Sciences | |
|---|---|
| **Field** | **Subfield** |
| Energy Science and Technology | D73: Energy Chemistry (72) |
| | D74: Energy Computing and Measurement (80) |
| | D75: Energy Storage Technology (80) |
| | D76: Energy-Saving Technology (80) |
| Nuclear Science and Technology | D77: Nuclear Detection Technology and Nuclear Electronics (70) |
| | D78: Radiometric Metrology (70) |
| | D79: Nuclear Instruments and Equipment (78) |
| | D80: Nuclear Materials and Process Technology (70) |
| | D81: Particle Accelerators (70) |
| | D82: Fission Reactor Engineering Technology (70) |
| | D83: Nuclear Fusion Engineering Technology (80) |
| | D84: Nuclear Power Engineering Technology (79) |
| | D85: Isotope Technology (95) |
| | D86: Nuclear Explosion Engineering (92) |
| | D87: Nuclear Safety (80) |
| | D88: Spent Fuel Reprocessing Technology (80) |
| | D89: Radiation Protection Technology (80) |
| | D90: Nuclear Facility Decommissioning Technology (80) |
| | D91: Radioactive Waste Treatment and Disposal Technology (80) |
| Electronics and Communication Technology | D92: Electronic Technology (736) |
| | D93: Information Processing Technology (27) |
| | D94: Communication Technology (50) |
| | D95: Optoelectronics and Laser Technology (81) |
| | D96: Semiconductor Technology (80) |
| | D97: Broadcasting and Television Engineering Technology (80) |
| | D98: Radar Engineering (80) |
| Computer Science and Technology | D99: Basic Disciplines of Computer Science and Technology (922) |
| | D100: Computer System Architecture (999) |
| | D101: Computer Software (228) |
| | D102: Computer Engineering (41) |
| | D103: Computer Applications (285) |

| Engineering and Technological Sciences | |
|---|---|
| **Field** | **Subfield** |
| | D104: Basic Disciplines of Chemical Engineering (64) |
| | D105: Chemical Measurement Technology and Instrumentation (80) |
| | D106: Chemical Transport Processes (80) |
| | D107: Chemical Separation Engineering (80) |
| | D108: Chemical Reaction Engineering (80) |
| | D109: Chemical Systems Engineering (80) |
| | D110: Chemical Machinery and Equipment (75) |
| | D111: Inorganic Chemical Engineering (74) |
| Chemical Engineering | D112: Organic Chemical Engineering (80) |
| | D113: Electrochemical Engineering (77) |
| | D114: Coal Chemical Engineering (79) |
| | D115: Petrochemical Engineering (79) |
| | D116: Natural Gas Chemical Engineering (80) |
| | D117: Fine Chemical Engineering (76) |
| | D118: Papermaking Technology (86) |
| | D119: Fur and Leather Engineering (83) |
| | D120: Pharmaceutical Engineering (127) |
| | D121: Biochemical Engineering (116) |
| Engineering and Technology Related to Product Applications | D122: Product-Specific Application Technology (21) |
| | D123: Instrumentation Technology (80) |
| | D124: Weapons Science and Technology (90) |
| | D125: Textile Materials (80) |
| | D126: Fiber Manufacturing Technology (80) |
| | D127: Textile Technology (80) |
| Textile Science and Technolog | D128: Dyeing and Finishing Technology (80) |
| | D129: Clothing Technology (80) |
| | D130: Textile Machinery and Equipment (80) |
| | D131: Basic Disciplines of Food Science and Technology (80) |
| | D132: Food Packaging and Storage (77) |
| | D133: Food Machinery (80) |
| Food Science and Technology | D134: Processing and Utilization of By-Products in Food Processing (80) |
| | D135: Food Industry Business Management (86) |
| | D136: Food Engineering and Grain and Oil Engineering (80) |

| Engineering and Technological Sciences | |
| --- | --- |
| **Field** | **Subfield** |
| | D137: History of Architecture (85) |
| | D138: Building Materials (175) |
| | D139: Civil and Architectural Structures (108) |
| | D140: Civil and Architectural Engineering Design (235) |
| Civil and Architectural Engineering | D141: Basic Disciplines of Civil and Architectural Engineering (80) |
| | D142: Civil and Architectural Engineering Surveying (80) |
| | D143: Engineering Structures (80) |
| | D144: Civil and Architectural Engineering Construction (80) |
| | D145: Civil Engineering Machinery and Equipment (80) |
| | D146: Municipal Engineering (80) |
| | D147: Architectural Economics (80) |
| | D148: Basic Disciplines of Hydraulic Engineering (173) |
| | D149: Hydraulic Engineering Surveying (70) |
| | D150: Hydraulic Materials (79) |
| | D151: Hydraulic Structures (80) |
| | D152: Hydraulic Machinery (74) |
| Hydraulic Engineering | D153: Hydraulic Engineering Construction (92) |
| | D154: River Sediment Engineering (85) |
| | D155: Environmental Hydraulics (96) |
| | D156: Water Resources Management (72) |
| | D157: Flood Control Engineering (78) |
| | D158: Hydraulic Economics (69) |
| | D159: Road Engineering (79) |
| | D160: Highway Transportation (76) |
| | D161: Railway Transportation (80) |
| Transportation Engineering | D162: Waterway Transportation (80) |
| | D163: Ship and Vessel Engineering (80) |
| | D164: Air Transportation (80) |
| | D165: Transportation Systems Engineering (80) |
| | D166: Transportation Safety Engineering (80) |
| | D167: Basic Disciplines of Aviation and Aerospace Science and Technology (80) |
| | D168: Aircraft Structure and Design (80) |
| | D169: Spacecraft Structure and Design (80) |
| | D170: Aviation and Aerospace Propulsion Systems (80) |

| Engineering and Technological Sciences | |
|---|---|
| **Field** | **Subfield** |
| Aviation and Aerospace Science and Technology | D171: Aircraft Instruments and Equipment (80) |
| | D172: Aircraft Control and Navigation Technology (78) |
| | D173: Aviation and Aerospace Materials (80) |
| | D174: Aircraft Manufacturing Technology (84) |
| | D175: Aircraft Testing Technology (80) |
| | D176: Aircraft Launch, Recovery, and Flight Technology (84) |
| | D177: Aviation and Aerospace Ground Facilities and Technical Support (79) |
| | D178: Aviation and Aerospace Systems Engineering (89) |
| Environmental Science and Technology and Resource Science and Technology | D179: Basic Disciplines of Environmental Science and Technology (203) |
| | D180: Environmental Science (138) |
| | D181: Environmental Engineering (493) |
| | D182: Resource Science and Technology (24) |
| Safety Science and Technology | D183: Public Safety (259) |
| | D184: Basic Disciplines of Safety Science and Technology (70) |
| | D185: Safety Social Science (75) |
| | D186: Safety Material Science (75) |
| | D187: Safety Ergonomics (83) |
| | D188: Safety Systems Science (82) |
| | D189: Safety Engineering Technology (78) |
| | D190: Safety and Health Engineering Technology (82) |
| | D191: Safety Social Engineering (83) |
| | D192: Sector-Specific Safety Engineering Theory (96) |
| Management Science | D193: History of Management Thought (84) |
| | D194: Management Theory (80) |
| | D195: Management Metrology (81) |
| | D196: Sector Economic Management (80) |
| | D197: Regional Economic Management (80) |
| | D198: Science and Technology Management (80) |
| | D199: Public Administration (80) |
| | D200: Human Resource Development and Management (80) |
| | D201: Futures Studies (80) |
| | D202: Enterprise Management (600) |
| | D203: Management Engineering (71) |

5. HUMANITIES AND SOCIAL SCIENCES

| Humanities and Social Sciences | |
|---|---|
| **Field** | **Subfield** |
| Marxism | E1: Studies on Marx, Engels, Lenin, and Stalin (103) |
| | E2: Scientific Socialism (88) |
| | E3: Foreign Marxism Studies (81) |
| | E4: Mao Zedong Thought Studies (888) |
| | E5: History of Marxist Thought (416) |
| | E6: History of Socialist Movements (104) |
| Philosophy | E7: Marxist Philosophy (769) |
| | E8: History of Chinese Philosophy (21) |
| | E9: History of Western Philosophy (548) |
| | E10: Modern Foreign Philosophy (1) |
| | E11: Logic (368) |
| | E12: Ethics (69) |
| | E13: Aesthetics (976) |
| Religious Studies | E14: Religious Theory (60) |
| | E15: Primitive Religions (80) |
| | E16: Ancient Religions (80) |
| | E17: Buddhism (70) |
| | E18: Christianity (74) |
| | E19: Islam (80) |
| | E20: Taoism (80) |
| | E21: Judaism (80) |
| | E22: Hinduism (80) |
| | E23: Zoroastrianism (80) |
| | E24: Manichaeism (80) |
| Linguistics | E25: General Linguistics (199) |
| | E26: Comparative Linguistics (44) |
| | E27: Linguistic Geography (26) |
| | E28: Sociolinguistics (86) |
| | E29: Psycholinguistics (52) |
| | E30: Applied Linguistics (861) |
| | E31: Chinese Language Studies (439) |
| | E32: Languages and Scripts of Chinese Ethnic Minorities (24) |
| | E33: Foreign Languages (202) |
| Literature | E34: Literary Theory (231) |
| | E35: Literary Aesthetics (99) |
| | E36: Literary Criticism (89) |
| | E37: Comparative Literature (81) |
| | E38: Modern Chinese Literature (80) |
| | E39: Ancient Chinese Literature (355) |
| | E40: Chinese Genre Literature (82) |
| | E41: Chinese Folklore Literature (80) |
| | E42: Literature of Chinese Ethnic Minorities (80) |
| | E43: World Literature History (80) |
| | E44: Eastern Literature (80) |

| Humanities and Social Sciences | |
|---|---|
| **Field** | **Subfield** |
| | E45: Russian Literature (80) |
| | E46: Chinese Children's Literature (390) |
| | E47: British Literature (81) |
| | E48: French Literature (81) |
| | E49: German Literature (21) |
| | E50: Art Psychology (82) |
| | E51: Music (36) |
| | E52: Drama (45) |
| | E53: Traditional Chinese Opera (31) |
| | E54: Dance (30) |
| Art Studies | E55: Film (29) |
| | E56: Radio and Television Arts (21) |
| | E57: Fine Arts (869) |
| | E58: Applied Arts (46) |
| | E59: Calligraphy (26) |
| | E60: Photography (27) |
| | E61: Ancient Chinese History (66) |
| | E62: World General History (82) |
| | E63: Asian History (76) |
| History | E64: African History (21) |
| | E65: European History (87) |
| | E66: Historiography Theory (80) |
| | E67: Historical Documentation (72) |
| | E68: General Chinese History (80) |
| | E69: Archaeological Theory (81) |
| | E70: History of Archaeology (80) |
| | E71: Archaeological Technology (80) |
| Archaeology | E72: Chinese Archaeology (26) |
| | E73: Foreign Archaeology (30) |
| | E74: Specialized Archaeology (22) |
| | E75: Political Economics (21) |
| | E76: Economic Geography (29) |
| | E77: Developmental Economics (87) |
| | E78: Economic History (691) |
| | E79: World Economics (462) |
| | E80: Management Economics (21) |
| | E81: Accounting (718) |
| Economics | E82: Technical Economics (328) |
| | E83: Labor Economics (22) |
| | E84: Urban Economics (229) |
| | E85: Resource Economics (21) |
| | E86: Logistics Economics (644) |
| | E87: Commercial Economics (418) |
| | E88: Information Economics (544) |
| | E89: Public Finance (427) |
| | E90: Finance (404) |

| Humanities and Social Sciences | |
|---|---|
| **Field** | **Subfield** |
| Political Science | E91: Political Science Theory (303) |
| | E92: Political Systems (87) |
| | E93: Public Administration (398) |
| | E94: International Politics (84) |
| Law | E95: Theoretical Jurisprudence (376) |
| | E96: Legal History (155) |
| | E97: Sectoral Law (6471) |
| | E98: International Law (476) |
| Military Science | E99: Military Theory (80) |
| | E100: Military History (80) |
| | E101: Military Psychology (80) |
| | E102: Strategic Studies (80) |
| | E103: Operational Studies (80) |
| | E104: Tactical Studies (80) |
| | E105: Military Command Studies (80) |
| | E106: Military Organization Studies (80) |
| | E107: Military Political Work Studies (80) |
| | E108: Military Logistics (80) |
| | E109: Military Geography (80) |
| | E110: Military Technology (80) |
| Sociology | E111: History of Sociology (48) |
| | E112: Sociological Theory (1089) |
| | E113: Sociological Methods (324) |
| | E114: Experimental Sociology (21) |
| | E115: Applied Sociology (1016) |
| | E116: Social Geography (30) |
| | E117: Cultural Sociology (45) |
| | E118: Economic Sociology (56) |
| | E119: Social Anthropology (63) |
| | E120: Organizational Sociology (168) |
| | E121: Developmental Sociology (34) |
| | E122: Welfare Sociology (115) |
| | E123: Demography (8) |
| | E124: Labor Science (29) |
| Ethnology and Cultural Studies | E125: Cultural Anthropology and Folklore (79) |
| | E126: Cultural Studies (86) |
| | E127: Tibetology (95) |
| | E128: Xinjiang Ethnic Studies (85) |
| | E129: World Ethnic Studies (47) |
| Journalism and Communication Studies | E130: Journalism Theory (170) |
| | E131: History of Journalism (872) |
| | E132: Journalism Practice (35) |
| | E133: Journalism Business Management (92) |
| | E134: Radio and Television (81) |
| | E135: Communication Studies (458) |
| | E136: Journalism Operations (80) |

| Humanities and Social Sciences | |
|---|---|
| **Field** | **Subfield** |
| Education | E137: History of Education (592) |
| | E138: Principles of Education (82) |
| | E139: Teaching Methodology (56) |
| | E140: Moral Education Principles (590) |
| | E141: Educational Sociology (339) |
| | E142: Educational Management (26) |
| | E143: Educational Technology (2125) |
| | E144: General Education (277) |
| | E145: Vocational and Technical Education (34) |
| Sports Science | E146: Exercise Physiology (907) |
| | E147: History of Sports (86) |
| | E148: Sports Theory (80) |
| | E149: Sports Biomechanics (81) |
| | E150: Sports Psychology (80) |
| | E151: Sports Health Science (80) |
| | E152: Physical Education (80) |
| Statistics | E153: Economic Statistics (70) |
| | E154: Science and Technology Statistics (85) |
| | E155: Environmental and Ecological Statistics (80) |
| | E156: Biological and Medical Statistics (82) |
| | E157: Biological and Medical Statistics (82) |
| Library, Information, and Documentation | E158: Information Science (89) |
| | E159: Archival Science (52) |
| Science | E160: Museum Studies (112) |

