| mation, and Documenta- | E159: Archival Science (52) |
| tion | |
| Science | E160: Museum Studies (112) |

## E  EVALUATION METHOD AND DIFFICULTY LEVEL CALIBRATION

Below we provide the detailed methodology for model evaluation and the calibration process used to assign difficulty levels to the EESE instances, addressing specific reviewer queries.

### E.1  MODEL EVALUATION METHOD

The evaluation process in this study follows the "LLM-as-a-judge" paradigm. Specifically, we first present questions to the model under test and record its responses. These responses, along with the ground-truth answers, are then provided to the judge model GPT-4o, which is explicitly informed of the question type (objective or subjective). For objective questions, a binary scoring criterion is applied, where a correct answer receives 10 points and an incorrect answer receives 0. For subjective questions, the judge model assigns a continuous score between 0 and 10 based on response quality. The scores for all questions are subsequently averaged and normalized to a percentage scale to represent the model's overall performance.

### E.2  DIFFICULTY LEVEL CALIBRATION

To assess the difficulty level of questions in EESE-Pool, we selected 6 representative models (including DeepSeek-R1, O3, GPT-4o, Grok-3, Gemini-2.5-pro, Qwen2.5-72B-Instruct) and prompted each to answer every question independently. Using GPT-4o as the judge model, we computed the average score achieved by these models on each question. Based on this mean score, questions are classified into three difficulty tiers: those with a score below 4 are labeled as "Hard", scores from 4 to 7 (inclusive) as Middle", and scores exceeding 7 as "Easy".

# F HUMAN EXPERT RECRUITMENT AND INVOLVEMENT

This appendix details the protocols for recruiting human experts and their involvement in the construction of the EESE benchmark, covering recruitment, task specifications, compensation, and ethical considerations.

## F.1 RECRUITMENT AND QUALIFICATION

**Recruitment Method:** Experts were primarily recruited through our academic collaboration networks, targeting top-tier universities and research institutions to ensure high levels of expertise and reliability.

**Qualification Requirements:** All recruited experts were required to hold a Master's or Ph.D. degree in a relevant scientific discipline, or to possess several years of high-level teaching or research experience.

## F.2 TASK SPECIFICATIONS AND COMPETENCY REQUIREMENTS

Expert involvement was structured across different stages of data construction and refinement, with tasks categorized by required cognitive load and expertise.

### F.2.1 DATA ENGINE

**Transcription** Experts collected instances from textbooks, question banks, and online resources, transcribing them into a standardized format. This task required foundational domain knowledge and strict adherence to data formatting and fact-checking protocols.

**Expansion** Experts contributed high-value instances for uncovered subfields by synthesizing field knowledge, practical experience, and pedagogical insights. This demanded high professional competence, with a focus on addressing knowledge gaps and ensuring instance novelty.

**Coarse-grained Quality Control** Experts reviewed and manually modified instances flagged by LLMs for errors in formatting, factual accuracy, or logical coherence. This required critical thinking and a mastery of LLM error review guidelines and difficulty verification standards.

### F.2.2 DATA REFINEMENT

**Enhancement By Distraction (Low HI)** Experts verified the correctness and relevance of auto-generated distractors, serving as the final fine-grained quality control step. This required **familiarity with subject matter** to ensure distractors were highly discriminative without introducing factual errors.

**Enrichment By Cross-Disciplinary (Medium HI)** Experts conducted a fine-grained review and refinement of interdisciplinary content to ensure factual precision and educational alignment. This demanded the ability to integrate knowledge across disciplines and design complex, yet coherent, interdisciplinary links.

**Expert-Driven Refinement (High HI)** Experts manually rewrote or restructured problems to enhance clarity, embed subtle complexity, or decompose multi-step reasoning, followed by fine-grained quality validation. This task had the highest level of expertise requirement, necessitating the ability to reconstruct complex logical structures and perform meticulous quality validation.

## F.3 COMPENSATION STRUCTURE

To ensure the scientific rigor of the EESE benchmark, significant resources were allocated for human expertise. A total of 609 experts contributed to the construction of the 100K+ instance EESE-Pool.

**Total Investment:** The total cost for the entire construction process was approximately $428,057.77. The cumulative effort amounted to approximately 30,510 expert-hours. [1]

**Compensation Strategy:** A strict tiered compensation strategy was implemented to align remuneration with the professionalism and complexity of each task. Basic tasks, such as *Transcription* and *Coarse-grained QC*, were compensated at a standard rate. Core tasks demanding high expertise and cognitive load, such as *Expansion* and *Expert-Driven Refinement (High HI)*, were compensated at a premium rate, with a maximum hourly rate of $45. The average hourly rate across the project was approximately $14.03.

## F.4 INFORMED CONSENT

Prior to their involvement, all experts were required to sign an Informed Consent form and a service agreement. These documents clearly outlined the nature of their tasks, the scope of data usage, and the compensation structure. This process ensured that all experts participated in the EESE construction on a fully informed and voluntary basis.

---

[1] All monetary and hour amounts here are approximate and not exact financial figures.