# OpenReview forum: "The Ever-Evolving Science Exam"
_ICLR.cc/2026/Conference — ICLR 2026 Conference Withdrawn Submission_

### Official Review · Reviewer_oX8M · 2025-10-23

**Soundness:** 3
**Presentation:** 2
**Contribution:** 2
**Rating:** 4
**Confidence:** 4

**Summary:**

**Motivation**: The paper aims to provide a science benchmark that is robust to data leakage and cheaper to run than very large static suites. The authors argue that public benchmarks risk leakage and that exhaustive evaluation is costly.

**Approach**: They create a large non-public pool of more than 100k curated question answer items across five disciplines and 500 plus subfields, then periodically resample a 500 item subset named EESE for evaluation. The pool is built with a three stage pipeline and a three branch refinement process to increase difficulty. The subset is refreshed over time to reduce leakage and compute cost.

**Key Results**: They evaluate 32 models and report that thinking style models tend to outperform non thinking ones, that humans remain well ahead, that discipline specific gaps are large, and that the small resampled EESE tracks the larger pool by rank correlation while keeping cost lower. They also show that refinement lowers model accuracy, which they present as evidence that difficulty increased.

**Strengths:**

**Usefulness**: A periodically refreshed 500 item suite can lower evaluation cost while still surfacing rank differences among models. The reported rank correlations between the 500 item subset and the full pool suggest the subset is a reasonable proxy for model ordering. This can help labs iterate faster on science capabilities while avoiding full scale runs.

**Human curated difficulty**: The multi stage refinement, including expert driven edits and targeted distractor design, plausibly increases difficulty and reduces trivial items. The authors show consistent drops in accuracy after refinement, which supports the claim that questions became harder. Expert involvement likely improves clarity and reduces labeling errors, which is valuable for fair evaluation.

**Coverage**: The pool spans many subfields and formats, which can capture different skills such as recall, selection with distractors, and limited open ended reasoning. This breadth improves the chance that weaknesses in particular scientific areas will be exposed.

**Weaknesses:**

**Comparability over time**: Because each release is a fresh resample, it is hard to compare new model results to older baselines or to chart model progress across time stamps. Without anchors or versioned fixed sets, changes in scores may reflect sampling noise as much as model improvement. This weakens the utility for longitudinal tracking and for comparing to external baselines that were reported on a different draw.

**Novelty**: There are many science benchmarks and pools that use transcription, expert curation, and difficulty control. The main novelty claimed here is the resampling based leakage mitigation. That choice also introduces cost and comparability challenges since many baselines must be re run whenever the subset changes. The net novelty feels modest relative to prior benchmark design patterns.

**Reproducibility**: The paper calls the pool non public while the reproducibility section states that code and datasets will be made publicly available. This needs to be reconciled. If the 100k plus pool stays private, then full reproduction and community baseline reuse are limited. If it becomes public, the leakage claim becomes weaker. Clarifying what will be released and when is important.

**Presentation quality**: Table text is small and hard to read in places such as Table 1. Figure 5 appears to lack some labeling details and the visual style of several figures feels closer to promotional animations than archival scientific figures. These choices hurt readability.

**Terminology misuse**: Using the word “evolving” suggests regeneration or distributional shift over time. The method is resampling from a fixed non public pool, not generating new problems. “Resampled” or “rotating” would be more precise and would avoid confusion.

**Questions:**

1. Do I have to rerun every baseline each time the benchmark is resampled. Is there any efficient workaround?
2. Can the authors quantify the magnitude of sampling variance in EESE and set reporting rules?
3. Can sampling 500 samples capture enough coverage of diverse scientific questions?

---

> ### Author Response · Authors · 2025-11-27
> **Official Responses to Reviewer oX8M**
>
> 1. **Clarity of comparability over time in response to W1**
>
>    ````
>    Comparability over time: Because each release is a fresh resample, it is hard to compare new model results to older baselines or to chart model progress across time stamps. Without anchors or versioned fixed sets, changes in scores may reflect sampling noise as much as model improvement. This weakens the utility for longitudinal tracking and for comparing to external baselines that were reported on a different draw.
>    ````
>
>    **Response:**
>
>    We appreciate this insightful point. We address the challenge of sampling noise and baseline tracking through a three-layered approach:
>
>    1) Statistical Alignment: We strictly control sampling noise via Stratified Sampling. By enforcing identical distributions of difficulty levels and disciplines across releases, we ensure that each new subset is statistically equivalent to previous ones. This design minimizes inherent noise before evaluation begins.
>
>    2) Anchor Models: To replace static question anchors (which risk leakage), we utilize Anchor Models (e.g., fixed versions of open-source models) as stable references. We re-evaluate these anchors on each new release. Consistent performance from these stable models serves as a empirical validation that the benchmark's difficulty remains constant over time.
>
>    3) Versioned Snapshots: For exact comparisons against external baselines reported on different draws, we maintain a permanent archive of all Versioned Snapshots. Researchers can access any legacy dataset to reproduce historical results exactly or to conduct longitudinal tracking on a fixed set if preferred.
>
>
>
> 2. **Clarification of novelty in response to W2**
>
>    ````
>    Novelty: There are many science benchmarks and pools that use transcription, expert curation, and difficulty control. The main novelty claimed here is the resampling based leakage mitigation. That choice also introduces cost and comparability challenges since many baselines must be re run whenever the subset changes. The net novelty feels modest relative to prior benchmark design patterns.
>    ````
>
>    **Response:**
>
>    We clarify that EESE introduces a new benchmarking paradigm, shifting from static artifacts to a dynamic evaluation ecosystem. Our core contribution is the 100K+ expert-refined EESE-Pool, which serves as the necessary infrastructure to resolve data leakage in existing benchmarks. This dynamic system represents a fundamental advancement over prior designs, offering the first sustainable, leakage-resilient solution for long-term scientific evaluation.
>
>    Regarding cost and comparability, our approach actually reduces the community burden. Evaluating a 500-instance proxy incurs negligible computational cost compared to re-evaluating massive static datasets. We ensure comparability via stratified sampling, where each release rigorously aligns with the source pool's distribution. As evidenced in Figure 6(b), these subsets serve as stable statistical proxies, ensuring valid cross-version comparisons. Thus, EESE provides a robust, forward-compatible standard that effectively differentiates model capabilities where static benchmarks now fail.
>
>
>
> 3. **Clarity of reproducibility and data release in response to W3**
>
>    ````
>    Reproducibility: The paper calls the pool non public while the reproducibility section states that code and datasets will be made publicly available. This needs to be reconciled. If the 100k plus pool stays private, then full reproduction and community baseline reuse are limited. If it becomes public, the leakage claim becomes weaker. Clarifying what will be released and when is important.
>    ````
>
>    **Response:**
>
>    We clarify that the "publicly available" guarantee refers specifically to the EESE evaluation subsets used to generate our results. To reconcile community access with leakage resistance, we implement a "Controlled Gradual Release" strategy. Specifically, we release fully open-source batches of 500 questions every quarter. To date, we have released two versions (1,000 questions), with the ultimate goal of making the full 100K+ pool public in stages over time.
>
>    This approach resolves the trade-off between reproduction and leakage. For reproduction, we employ a "Snapshot Version System" to permanently archive every released batch, ensuring the specific data underpinning Table 1 remains accessible for verification. For leakage, the controlled release rate ensures the vast majority of the reserve remains unseen by models during training. This strategy balances the scientific requirement for open verifiability with the practical necessity of maintaining a leakage-resilient benchmark.

---

> ### Author Response · Authors · 2025-11-27
> **Official Responses to Reviewer oX8M**
>
> 4. **Improvement of presentation quality in response to W4**
>
>    ````
>    Presentation quality: Table text is small and hard to read in places such as Table 1. Figure 5 appears to lack some labeling details and the visual style of several figures feels closer to promotional animations than archival scientific figures. These choices hurt readability.
>    ````
>
>    **Response:**
>
>    We sincerely appreciate the feedback.
>
>    We will comprehensively overhaul the visualization strategy in the revision. Concretely, we will redesign Table 1 by increasing the font size and optimizing the layout to ensure maximum legibility. For Figure 5 and other diagrams, we will shift from an illustrative aesthetic to a standard, schematic scientific style, removing extraneous artistic elements while adding precise technical labeling.
>
>
>
> 5. **Precision of terminology in response to W5**
>
>    ````
>    Terminology misuse: Using the word “evolving” suggests regeneration or distributional shift over time. The method is resampling from a fixed non public pool, not generating new problems. “Resampled” or “rotating” would be more precise and would avoid confusion.
>    ````
>
>    **Response:**
>
>    We sincerely appreciate the suggestions. While we agree that 'Resampling' accurately describes the extraction mechanism, we chose 'Evolving' to capture the broader, dynamic lifecycle of the benchmark system as shown in Figure 2. Unlike static datasets, EESE is 'Evolving' in three dimensions:
>
>    1) Growth: The Data Engine allows the source pool to expand continuously (Phase II: Expansion) to cover new scientific subfields.
>    2) Adaptation: The benchmark can shift its difficulty distribution (via the Refinement Framework in Figure 3) to co-evolve with rising model capabilities, ensuring it never hits a saturation ceiling.
>    3) Self-Correction: Community feedback on released slices triggers iterative refinement of the underlying pool.
>
>    Thus, 'Evolving' refers to the system's capability to survive, grow, and adapt, not just the rotation of fixed content.
>
>
>
> 6. **Efficiency of evaluation in response to Q1**
>
>    ````
>    Do I have to rerun every baseline each time the benchmark is resampled. Is there any efficient workaround?
>    ````
>
>    **Response:**
>
>    Thank for your question.
>
>    Rerunning baselines is unnecessary. We employ a "Pre-computation & Retrieval" strategy to eliminate this burden. Since we conduct comprehensive internal evaluations of 32 leading models across the entire EESE-Pool, valid performance logs for all instances already exist. Upon each resampling, we simply retrieve and publish the pre-computed scores of these baselines on the new 500-instance subset. Consequently, the community incurs zero cost for baseline updates. Users only need to evaluate their own target models on the small 500-instance proxy, which requires negligible computation (minutes) compared to traditional massive benchmarks.
>
>
>
> 7. **Quantification of sampling variance in response to Q2**
>
>    ````
>    Can the authors quantify the magnitude of sampling variance in EESE and set reporting rules?
>    ````
>
>    **Response:**
>
>    Thank for your question. We quantify the sampling variance through rigorous empirical simulation and establish clear reporting guidelines based on these results.
>
>    1) Quantification of Variance. We rigorously validate stability by performing 100 iterations of stratified sampling from the EESE-Pool. As shown in the tables below, the performance of representative models remains highly consistent across different draws. The variance is negligible (σ²≈0.0001), which confirms that our stratified strategy effectively suppresses sampling noise.
>
>    **Table: Model performance across selected sampling iterations.**
>
>    | Model                | Sam.1  | Sam.5  | Sam.10 | Sam.20 | Sam.30 | Sam.50 | Sam.80 | Sam.100 |
>    | :------------------- | :----- | :----- | :----- | :----- | :----- | :----- | :----- | :------ |
>    | O3                   | 0.4218 | 0.3935 | 0.4038 | 0.4184 | 0.4015 | 0.4072 | 0.4032 | 0.3922  |
>    | Grok-4               | 0.3458 | 0.3493 | 0.3397 | 0.3361 | 0.3226 | 0.3355 | 0.3377 | 0.352   |
>    | Deepseek-V3          | 0.2536 | 0.247  | 0.261  | 0.2604 | 0.2526 | 0.2467 | 0.2571 | 0.2655  |
>    | GPT-4o               | 0.2415 | 0.2423 | 0.2502 | 0.2361 | 0.243  | 0.2392 | 0.2432 | 0.2346  |
>    | Qwen2.5-72B-Instruct | 0.2039 | 0.195  | 0.19   | 0.2007 | 0.1894 | 0.1997 | 0.1879 | 0.1998  |
>
>    **Table: Overall mean and variance across 100 samplings.**
>
>    | Model                | Variance | Mean Performance |
>    | :------------------- | :------- | :--------------- |
>    | O3                   | 0.000211 | 0.409            |
>    | Grok-4               | 0.000106 | 0.3418           |
>    | Deepseek-V3          | 0.000086 | 0.2498           |
>    | GPT-4o               | 0.000079 | 0.2412           |
>    | Qwen2.5-72B-Instruct | 0.000076 | 0.1921           |

---

> ### Author Response · Authors · 2025-11-27
> **Official Responses to Reviewer oX8M**
>
> 7. **Quantification of sampling variance in response to Q2  (continued)**
>
>
>    2) Reporting Rules. Based on this quantification, we define the following reporting standards for EESE users: Version Citation: Users must report the specific Snapshot Version ID (e.g., EESE-v1, EESE-v2) used for evaluation. Significance Threshold: Given that the empirical standard deviation is approximately 1.0-1.4% (Variance), we advise that performance gaps smaller than 1% should be interpreted as within the margin of error, whereas gaps exceeding this threshold indicate significant model differences. We will include these guidelines in the benchmark documentation.
>
>
>
> 8. **Sufficiency of coverage in response to Q3**
>
>    ````
>    Can sampling 500 samples capture enough coverage of diverse scientific questions?
>    ````
>
>    **Response:**
>
>    Thank for your question.
>
>    We confirm that the 500-sample size is sufficient to ensure both broad disciplinary coverage and statistical representativeness, thanks to our combined prior–posterior sampling paradigm.
>
>    On the prior side, we employ a strict stratified sampling strategy based on the hierarchical taxonomy of the 100K+ EESE-Pool. By strictly aligning the sample proportions with the discipline and subfield distribution of the full pool, we structurally guarantee that diverse scientific domains (from fundamental sciences to specialized interdisciplinary fields) are adequately represented, preventing the underrepresentation of long-tail subjects.
>
>    On the posterior side, empirical validation confirms that this subset serves as a reliable proxy for the full benchmark. As illustrated in Figure 6, the performance rankings of models evaluated on the 500-instance subset demonstrate a high Spearman Rank Correlation Coefficient (SRCC) with their rankings on the full 100K-item dataset. This strong statistical alignment indicates that the 500-instance subset preserves the discriminative characteristics and difficulty distribution of the larger pool.

---

### Official Review · Reviewer_JFgV · 2025-10-31

**Soundness:** 3
**Presentation:** 3
**Contribution:** 4
**Rating:** 6
**Confidence:** 3

**Summary:**

The paper presents the Ever-Evolving Science Exam (EESE), a benchmark consisting of over 100 000 questions and answer pairs from five science disciplines. The benchmark is designed so that it will sample 500 questions (EESE) from the total dataset (EESE-Pool) and benchmark LLMs on this subset. The subset of 500 questions will be resampled from time to time to ensure that there is little data-leakage from the datasets to the LLMs to ensure that the benchmark stays relevant over time.

The dataset has a broad range (100k questions), wide reach (5 disciplines, 500 subfields, wide range of question formats) and high rigor (systematic and principled process to ensure quality).

Finally, the dataset is used for a comprehensive evaluation of LLMS, benchmarking 32 different models, including thinking models one both EESE-Pool and EESE. The benchmark reveals that human experts outperform the best LLMs and shows that the benchmark is demanding for current LLMs.

**Strengths:**

* The paper is easy to read and understand; it is well-written and structured well.
* Comprehensive dataset of 100k question answer pairs with a nice evaluation set of 500 samples (stratified sampling across difficulty levels) that will be resampled from time to time.
* Rigorous process for quality checking and human involvement.
* Huge evaluating on 32 different LMMs models.
* The results are clear and the discussion nice.

**Weaknesses:**

* The paper leaves a lot of detail out, so my understanding of it is only partial.
* The description of the dataset and what is measures is not clear from time to time. I disagree that the benchmark assesses scientific capabilities, as is stated in the abstract. My understanding is that this benchmark is a test of knowledge about science and not scientific capabilities. Similar examples of imprecise language about what the benchmark tests exist elsewhere in the text as well.
* I am not really a fan of Figure 2. I find its “the artistic form” to reduce the clarity, and I would prefer that the process was illustrated with boxes and icons only. Figure 1 is better, as the artistic illustration does not hamper the readability.
* Limitations are not discussed. Please do.

Overall, this could be a very good addition to the literature. By clarifying the questions I have below, the presentation and soundness could improve, and my recommendation could improve with them.

**Questions:**

* Is the exam ever evolving? Resampling 500 question answering pairs from a set of 100 000 is not ever evolving. After some time (depending on resampling rate) the whole dataset will have been exposed. This could of course take a long time if the evaluation set changes every month, but it is not ever evolving …
* I wonder if data leakage has happened already. All 100k questions are asked to the best performing LLMs 1) during Categorization and 2) when the EESE-Pool performance was calculated. Given that the top-performing commercial models gather data on questions and answers, they have seen all questions already. This means that it is possible for them to improve their answers to these questions. Am I wrong?
* I do not understand the sentence that starts on line 183: “EESE is then randomly …”. Is this an ongoing process or does it happen as part of something else?
* Line 242: What is meant by course-grained control?
* It is not clear how the predefined thresholds for the three difficulty levels are decided. Please specify.
* Also, are outlier cases not used for setting the thresholds?
* It is not clear how the stratified sampling of difficulty levels is done, nor how many questions of each level a 500 sample consists of. Could you please explain?
* Is the Three-Branch Refinement Framework parallel? Do questions go through all three branches at the same time, or will they only go through one branch? The text indicates the latter, but then I would not state that it is a parallel framework.
* Could you please be explicit about what “resilient against data leakage” means?
* It is not clear to me how the evaluation set EESE is shared with the world. Is it through an API?
* In the caption of Table 1, it is stated that third best is underlined as well as the second best. Is this correct?
* How are human experts found? Who are they? An what characteristics make them experts in this regard?
* Could you please explain SSH, AS, MS and so on. Please make them correspond to the terms in Figure 1.
* How is performance calculated? Number of correct answers divided by total of questions asked? This is not mentioned explicitly only implied. How is free text answers evaluated?
* In table 2, is not Performance a better term than Overall?
* On line 418: Is the word “cost” missing before $4.5\times$?
* Table 2 caption: Could you please state average of best models? Make it explicit.
* Given the results in Table 2, is not DeepSeek R1 the best compromise between cost and performance? Should this be mentioned?
* How are the results illustrated in Figure 6 b) made? Is it made from one sampling of the EESE-Pool or many?
* How can all relevant data and code be made publicly available, as stated in the reproducibility statement, without enabling the LLMs to optimize on it? It is also explicitly stated that the datasets used in the paper are publicly available. How can this be?
* Are model configurations and hardware details described in detail in the paper as stated in the reproducibility statement?

---

> ### Author Response · Authors · 2025-11-27
> **Official Responses to Reviewer JFgV**
>
> 1. **Clarification of "Detail Reconstruction" in response to W1**
>
>    ````
>    The paper leaves a lot of detail out, so my understanding of it is only partial.
>    ````
>
>    **Response:**
>
>    We greatly appreciate this feedback.
>
>    We would like to re-explain the core logic of EESE here. Specifically, EESE operates on a three-step principle: 1) **Construction:** A rigorously human-refined, 100K+ private pool serves as the foundation. 2) **Security**: The pool remains closed to prevent training data contamination (Leakage Resilience). 3) **Evaluation**: We quarterly release strictly versioned, public 500-instance subsets (Dynamic Sampling) to allow for community verification and model benchmarking without compromising the reserve.
>
>    Meanwhile, we have expanded **Appendix E** to detail the evaluation method and difficulty level calibration, and **Appendix F** to detail the recruitment and involvement of human experts in EESE-Pool construction.
>
>
>
> 2. **Clarification of "Scientific Capabilities" in response to W2**
>
>    ````
>    The description of the dataset and what is measures is not clear from time to time. I disagree that the benchmark assesses scientific capabilities, as is stated in the abstract. My understanding is that this benchmark is a test of knowledge about science and not scientific capabilities. Similar examples of imprecise language about what the benchmark tests exist elsewhere in the text as well.
>    ````
>
>    **Response:**
>
>    We appreciate the reviewer’s emphasis on terminological precision.
>
>    First, knowledge serves as the essential foundation for scientific capabilities [1, 2, 3], as it enables the identification of principles and interpretation of contexts. More importantly, solving our science questions requires the reasoning capabilities of models. As shown in Figure 3 and Appendix B, sovling the chemistry question (Medium HI) requiring the integration of oxidation states, periodic trends, and electronegativity. This multi-step knowledge reasoning and integration is the scientific reasoning skill we are examining.
>
>
>
> 3. **Improvement of Figure 2 in response to W3**
>
>    ````
>    I am not really a fan of Figure 2. I find its “the artistic form” to reduce the clarity, and I would prefer that the process was illustrated with boxes and icons only. Figure 1 is better, as the artistic illustration does not hamper the readability.
>    ````
>
>    **Response:**
>
>    We thank the reviewer for this valuable feedback. We understand that while artistic elements can be engaging, the primary function of Figure 2 is to precisely convey the technical details of our Data Engine and Refinement process. We agree that strictly prioritizing clarity is essential for this methodological overview. Therefore, we have redesigned Figure 2 as suggested:
>
>    1) Adopting a Schematic Approach: We have replaced the artistic/illustrative elements with a clean, structured layout using standard boxes and icons.
>
>    2) Enhancing Readability: The new diagram focuses exclusively on the logical flow between the Transcription, Expansion, and Categorization stages, ensuring that the input-output relationships and the refinement loops are unambiguous.
>
>    We will update this figure within the next version.
>
>
>
> 4. **Discussion of Limitations in response to W4**
>
>    ````
>    Limitations are not discussed. Please do.
>    ````
>
>    **Response:**
>
>    We sincerely appreciate the reviewer’s constructive suggestion. We have drafted a dedicated Limitations subsection, which will be included in the Conclusion section of the revised manuscript. The specific content we have added is as follows:
>
>    First, while EESE encompasses a broad spectrum of scientific disciplines, the current iteration primarily evaluates text-based reasoning capabilities. Real-world scientific problem-solving often necessitates interpreting multimodal data (e.g., molecular diagrams, circuit schematics, and geological maps). Integrating such multimodal instances remains a critical direction for our future expansion.
>
>    Second, regarding open-ended questions, while our 0-10 scoring metric effectively measures factual alignment with the ground truth, it does not strictly audit the step-by-step validity of the Chain-of-Thought (CoT). Consequently, a model could theoretically achieve a high score by covering key information points even if its intermediate reasoning contains logical gaps. Future iterations aim to address this by introducing process-level scoring to strictly evaluate the rigor of the derivation steps.

---

> ### Author Response · Authors · 2025-11-27
> **Official Responses to Reviewer JFgV**
>
> 5. **Clarification of "Ever-Evolving" in response to Q1**
>
>    ````
>    Is the exam ever evolving? Resampling 500 question answering pairs from a set of 100 000 is not ever evolving. After some time (depending on resampling rate) the whole dataset will have been exposed. This could of course take a long time if the evaluation set changes every month, but it is not ever evolving …
>    ````
>
>    **Response:**
>
>    We appreciate the reviewer’s insightful observation. We justify the "Ever-Evolving" terminology based on two key aspects.
>
>    We update 500 instances every quarter based on the massive scale of the 100K+ EESE-Pool . In the fast-paced context of AI research, this duration effectively supports long-term, leakage-resilient evaluation. Furthermore, given that SOTA models currently score about 0.4 (Table 1), the benchmark’s difficulty will remain relevant long before the pool risks saturation.
>
>    Crucially, "Ever-Evolving" refers to our methodology. As illustrated in Figure 2, EESE is built upon a scalable Data Engine and Refinement Framework. This infrastructure allows us to continuously generate, verify, and inject new high-quality instances.
>
>
>
> 6. **Clarification of Data Leakage in response to Q2**
>
>    ````
>    I wonder if data leakage has happened already. All 100k questions are asked to the best performing LLMs 1) during Categorization and 2) when the EESE-Pool performance was calculated. Given that the top-performing commercial models gather data on questions and answers, they have seen all questions already. This means that it is possible for them to improve their answers to these questions. Am I wrong?
>    ````
>
>    **Response:**
>
>    We thank the reviewer for raising this critical point. We understand the concern, and we would like to clarify that data leakage is prevented through two rigorous safeguards:
>
>    During both categorization and evaluation, we **never feed the correct answers (Ground Truth) to the models**. Without the ground truth or a reward signal, the models cannot perform supervised learning or reinforcement learning to "correct" or "improve" their specific answers to these questions. Meanwhile, during the data construction process, all questions adapted or generated by LLMs are subsequently reviewed, curated, and rewritten **by human experts**, which helps minimize the risk of data leakage.
>
>
>
> 7. **Clarification of "Ongoing Process" in response to Q3**
>
>    ````
>    I do not understand the sentence that starts on line 183: “EESE is then randomly …”. Is this an ongoing process or does it happen as part of something else?
>    ````
>
>    **Response:**
>
>    We thank the reviewer for pointing out the question. We confirm that the resample is indeed an ongoing, periodic process.
>
>    The EESE evaluation set is **periodically resampled** from the massive EESE-Pool (100k+ instances). In each update cycle, after the random sampling, the selected instances undergo an additional round of expert verification and refinement to ensure they are high-quality and error-free before being used for evaluation. This mechanism ensures the benchmark remains leakage-resistant and ever-evolving.
>
>
>
> 8. **Definition of "Coarse-Grained Control" in response to Q4**
>
>    ````
>    Line 242: What is meant by course-grained control?
>    ````
>
>    **Response:**
>
>    We apologize for the ambiguity. "Coarse-grained control" refers to the foundational quality assurance measures applied during the Data Engine phase. It denotes a broad-pass filter designed to eliminate fundamental defects to ensure basic validity, such as formatting errors and obvious factual inaccuracies. As described in Section 3.1 (lines 209–212), this is implemented by using LLMs to flag potential errors, followed by manual expert correction.

---

> ### Author Response · Authors · 2025-11-27
> **Official Responses to Reviewer JFgV**
>
> 9. **Details of Difficulty Calibration and Sampling in response to Q5-Q7**
>
>    ````
>    Q5: It is not clear how the predefined thresholds for the three difficulty levels are decided. Please specify.
>    Q6: Also, are outlier cases not used for setting the thresholds?
>    Q7: It is not clear how the stratified sampling of difficulty levels is done, nor how many questions of each level a 500 sample consists of. Could you please explain?
>    ````
>
>    **Response:**
>
>    We thank the reviewer for raising this critical point. We address the difficulty calibration, outlier handling, and sampling strategy together to demonstrate the consistency of our dataset construction.
>
>    1) Difficulty Thresholds and Outlier Handling: We determine difficulty levels based on the empirical performance of 6 representative LLMs, evaluated by GPT-4o on a 0–10 scale. We establish three thresholds: Hard (Mean Score < 4), Medium (4 ≤ Score ≤ 7), and Easy (Score > 7). **Outlier cases are not used to set these thresholds**. Instead, the automated scoring process flags outliers (e.g., high score variance or ambiguity). As detailed in Section 3.1, domain experts manually intervene to calibrate these flagged cases. We assign their final difficulty levels only after this expert verification.
>
>    2) Stratified Sampling and Composition: To construct the 500-instance EESE, we employ stratified sampling across both Discipline and Difficulty dimensions to mirror the distribution of the full EESE-Pool. The resulting set consists of approximately 30% Easy, 30% Medium, and 40% Hard instances. This composition ensures the benchmark differentiates top-tier models while remaining accessible. More details about difficulty categorization can be find in Appendix E.
>
>    3) Validation of Representativeness: We validate this sampling strategy via the Spearman Rank Correlation Coefficient (SRCC) analysis shown in Figure 6(b). The high correlation between model rankings on the sampled EESE and the full EESE-Pool confirms that the 500-instance subset effectively serves as a robust proxy, preserving the difficulty variance and discriminatory power of the massive pool.
>
>
>
> 10. **Clarification of "Parallel" Framework in response to Q8**
>
>    ````
>    Is the Three-Branch Refinement Framework parallel? Do questions go through all three branches at the same time, or will they only go through one branch? The text indicates the latter, but then I would not state that it is a parallel framework.
>    ````
>
>    **Response:**
>
>    We appreciate the reviewers' comment. We use the term "Parallel" to emphasize the architectural independence and non-sequential nature of the three refinement branches. Specifically, the three branches (Distraction, Cross-Disciplinary, and Expert-Driven) operate autonomously. Modifications in one branch do not require the output of another.
>
>
>
> 11. **Definition of "Resilient Against Data Leakage" in response to Q9**
>
>    ````
>    Could you please be explicit about what “resilient against data leakage” means?
>    ````
>
>    **Response:**
>
>    We thank the reviewer for highlighting this critical concept. In our framework, resilient against data leakage means that the benchmark maintains its **evaluative integrity** even if the publicly available test set is inadvertently included in a model's training data. Specifically, the vast majority of our data (the 100k+ EESE-Pool) remains non-public. We only release a fresh, unseen subset (500 instances) as the EESE evaluation set at any given time, ensuring the new evaluation score remains a valid measure of capability.
>
>
>
> 12. **Mechanism of Sharing EESE in response to Q10**
>
>    ````
>    It is not clear to me how the evaluation set EESE is shared with the world. Is it through an API?
>    ````
>
>    **Response:**
>
>    We appreciate the reviewers' comment. To clarify, the EESE evaluation set is not accessed through an API. Instead, we release the data directly as **downloadable open-source files** (in JSON format) via our GitHub repository and HuggingFace Repo.
>
>    To date, we have already successfully released two versions of the EESE evaluation set following this mechanism. For each release, we provide the full set of 500 instances, including both questions and ground-truth answers.
>
>
>
> 13. **Clarification of Table 1 Formatting in response to Q11**
>
>    ````
>    In the caption of Table 1, it is stated that third best is underlined as well as the second best. Is this correct?
>    ````
>
>    **Response:**
>
>    We confirm that the caption and the table formatting are correct. We chose to underline both the second and third-best results to visually distinguish the top-tier performers (the top 3) from the rest of the models.
>
>    Specifically, the highest score is highlighted in bold, while the runners-up (2nd and 3rd) are underlined to provide a clear view of the leading group in each discipline. We have double-checked Table 1 to ensure this formatting rule is applied consistently across all columns.

---

> ### Author Response · Authors · 2025-11-27
> **Official Responses to Reviewer JFgV**
>
> 14. **Details of Human Experts in response to Q12**
>
>    ````
>    How are human experts found? Who are they? An what characteristics make them experts in this regard?
>    ````
>
>    **Response:**
>
>    We thank the reviewer for this critical inquiry. The details of high-level expertise have been added in **Appendix F**.
>
>    1) Recruitment: We employ a targeted recruitment strategy, engaging 609 domain experts primarily through academic collaboration networks, top-tier universities, and research institutions. We strictly avoid anonymous crowdsourcing to ensure accountability.
>
>    2) Qualifications: To qualify, every expert holds at least a Master’s or Ph.D. degree in their respective scientific discipline, typically accompanied by high-level teaching or research experience.
>
>    3) Competency Requirements: We implement a tiered assignment system that matches expert characteristics to task complexity. While foundational tasks (e.g., Transcription) require solid domain knowledge, high-cognitive tasks (e.g., Expert-Driven Refinement) are assigned to senior experts capable of synthesizing cross-disciplinary knowledge and restructuring complex reasoning chains.
>
>    4) Scale & Investment: The construction of the EESE-Pool involves approximately 30,510 expert-hourswith a total cost of **~$428,057** spanning 30,510 expert-hours. We have also detailed the tiered compensation structure (up to $45/hour for high-complexity tasks) to incentivize rigorous quality control.
>
>
>
> 15. **Explanation of Acronyms in response to Q13**
>
>    ````
>    Could you please explain SSH, AS, MS and so on. Please make them correspond to the terms in Figure 1.
>    ````
>
>    **Response:**
>
>    We thank for the reviewer’s question. These acronyms correspond to the five disciplinary categories detailed in **Appendix C** of our manuscript. Specifically: NS: Natural Sciences, AS: Agricultural Sciences, MS: Medical Sciences, ETS: Engineering and Technological Sciences, and SSH: Social Sciences and Humanities.
>
>    Further, we will update Figure 1 to include these standard abbreviations alongside the full category names, and add a clear legend to the caption of Table 1.
>
>
>
> 16. **Clarification of Performance Metrics in response to Q14**
>
>    ````
>    How is performance calculated? Number of correct answers divided by total of questions asked? This is not mentioned explicitly only implied. How is free text answers evaluated?
>    ````
>
>    **Response:**
>
>    We appreciate the reviewer’s comment. Here we provide a explicit clarification regarding the scoring metrics. Our evaluation uses GPT-4o as the judge, applying a binary score for objective questions and a continuous score for subjective ones. Please see Appendix E for more details.
>
>
>
> 17. **Refinements to Table 2 in response to Q15-Q18**
>
>    ````
>    Q15: In table 2, is not Performance a better term than Overall?
>    Q16: On line 418: Is the word “cost” missing before ?
>    Q17: Table 2 caption: Could you please state average of best models? Make it explicit.
>    Q18: Given the results in Table 2, is not DeepSeek R1 the best compromise between cost and performance? Should this be mentioned?
>    ````
>
>    **Response:**
>
>    We thank the reviewer for the detailed analysis of Table 2.
>
>    We will update the term "Performance" in column header of Table 2. We will revise the caption, including adding term "cost", and revise the term "average of best models" to explicitly list the specific models (i.e., Claude-3-7-sonnet, Deepseek-V3, and GPT-4.1). In addition, DeepSeek-R1 delivers strong reasoning performance significantly above standard models, yet its cost remains exceptionally low-comparable to, or even cheaper than, models without thinking. And we will add a specific discussion about this in Appendix section.
>
>
>
> 18. **Methodology for Figure 6(b) in response to Q19**
>
>    ````
>    How are the results illustrated in Figure 6 b) made? Is it made from one sampling of the EESE-Pool or many?
>    ````
>
>    **Response:**
>
>    We appreciate the reviewer’s comment.
>
>    Figure 6(b) presents the discipline correlations between EESE and EESE-Pool. To generate this result, we rank 32 models based on their performance on both the full EESE-Pool (100k+ instances) and the sampled EESE. The figure visualizes the Spearman Rank Correlation Coefficient (SRCC) computed between these two sets of rankings, defined as:
>    $$
>    \rho = 1 - \frac{6 \sum d_i^2}{n(n^2 - 1)}
>    $$
>    where $d_i$ represents the difference in ranks for each model and $n$ is the number of models. The consistently high correlation demonstrates that our stratified sampling strategy successfully preserves the statistical properties of the parent pool, confirming that EESE serves as a reliable and efficient proxy for large-scale evaluation.

---

> ### Author Response · Authors · 2025-11-27
> **Official Responses to Reviewer JFgV**
>
> 19. **Reconciling Public Availability with Optimization in response to Q20**
>
>    ````
>    How can all relevant data and code be made publicly available, as stated in the reproducibility statement, without enabling the LLMs to optimize on it? It is also explicitly stated that the datasets used in the paper are publicly available. How can this be?
>    ````
>
>    **Response:**
>
>    We thank the reviewer for highlighting this ambiguity. We will revise the manuscript to clarify our tiered release strategy, which balances reproducibility with the need to prevent data leakage:
>
>    1) Code and Methodology (Fully Public): All codes/methods regarding the Data Engine and evaluation pipelines is publicly available. This ensures that the community can reproduce our data construction process and methodology.
>
>    2) EESE Evaluation Subset (Public): The dynamically sampled 500-instance EESE set is released publicly. This allows researchers to inspect data format, verify difficulty, and validate results on the specific test set.
>
>    3) EESE-Pool (Held-Out): The full EESE-Pool is maintained as a private, held-out repository.
>
>
>
> 20. **Reproducibility of Model Configurations in response to Q21**
>
>    ````
>    Are model configurations and hardware details described in detail in the paper as stated in the reproducibility statement?
>    ````
>
>    **Response:**
>
>    We thank the reviewer for this important validity check. We will add a dedicated Implementation Details section in Appendix section.
>
>    Specifically, we have clarified the following. All evaluations are conducted in a zero-shot setting (Temperature = 0.0). Models such as Llama-3 and DeepSeek are evaluated using NVIDIA A100 (80GB) GPUs. For proprietary models (e.g., GPT-4o, Claude-3.5), we utilize official APIs and will make the specific versions dates used during testing public.
>
>
>
> **References:**
>
> [1] Bransford et al. How people learn: Brain, mind, experience, and school. National Academy Press. 1999.
>
> [2] Bao, Lei, et al. "Learning of content knowledge and development of scientific reasoning ability: A cross culture comparison." American journal of physics 77.12 (2009): 1118-1123.
>
> [3] Edelsbrunner et al. "The relation of representational competence and conceptual knowledge in female and male undergraduates." International Journal of STEM Education 10.1. 2023.

---

### Official Review · Reviewer_fMyi · 2025-10-31

**Soundness:** 2
**Presentation:** 2
**Contribution:** 2
**Rating:** 4
**Confidence:** 4

**Summary:**

This paper presents The Ever-Evolving Science Exam (EESE), a dynamic benchmark designed to evaluate the scientific reasoning capabilities of large foundation models while addressing two key issues with existing science benchmarks: data leakage and evaluation inefficiency. EESE consists of a large, non-public EESE-Pool (100K+ science question-answer pairs across 500+ subfields and five major disciplines) and a smaller, dynamic subset of 500 questions that are periodically resampled for evaluation. The benchmark emphasizes three principles—Range, Reach, and Rigor—supported by a multi-stage data construction process (Transcription, Expansion, Categorization) and a three-branch refinement pipeline aimed at increasing question difficulty and diversity. The paper evaluates 32 large models, both open and proprietary, and shows that EESE can differentiate scientific strengths and weaknesses across domains.

**Strengths:**

1. Ambitious and large-scale effort to build a benchmark addressing key issues of leakage and scalability.

2. Strong organization with clear design principles and methodology.

3. Inclusion of both natural and social science disciplines broadens scope beyond typical benchmarks.

4. Demonstrates clear model differentiation and human-model performance gaps, suggesting benchmark difficulty is appropriate.

5. Introduces a refinement process that systematically increases question complexity.

**Weaknesses:**

1. The dynamic “ever-evolving” aspect is underspecified. It is unclear how sustainable or reproducible the periodic updates are in practice.

2. Heavy reliance on proprietary models for evaluation and quality control (e.g., “thinking models”) makes reproducibility questionable. This also raises a question about how users can evaluate it without access to such models.

3. The use of LLMs to label and refine data raises circularity concerns and potential bias—models are used to create and test themselves.

4. The writing is verbose and could be more critical and analytical. The paper reads like an extensive benchmark manual rather than a scientific paper.

5. The evaluation mainly reports aggregate performance; there’s little qualitative analysis of what kinds of reasoning or question types models fail at.

6. No discussion of potential biases or limitations in expert selection, data sources, or field balance.

**Questions:**

1. How frequently will EESE be updated in practice, and what infrastructure or governance ensures that these updates are sustainable?

2. How do you ensure that using LLMs to annotate or refine instances does not bias the evaluation toward their own reasoning styles?

3. Could you provide more transparency on the cost and human involvement behind EESE’s refinement stages?

4. Is there a plan to make the dataset partially accessible (e.g., for academic verification) without compromising leakage resistance?

5. How does EESE compare in calibration stability to recent leakage-aware benchmarks such as AntiLeakBench or LessLeakBench?

---

> ### Author Response · Authors · 2025-11-27
> **Official Responses to Reviewer fMyi**
>
> 1. **Clarity of "Ever-Evolving" mechanism in response to W1 & Q1**
>
>    ````
>    W1: The dynamic “ever-evolving” aspect is underspecified. It is unclear how sustainable or reproducible the periodic updates are in practice.
>    Q1: How frequently will EESE be updated in practice, and what infrastructure or governance ensures that these updates are sustainable?
>    ````
>
>    **Response:**
>
>    We appreciate the questions concerning the operational details of our dynamic mechanism. We update EESE on a **quarterly basis** (every three months). To ensure practicality, we implement a rigorous protocol supported by robust infrastructure and strict governance:
>
>    1) Sustainability & Infrastructure: We rely on the massive scale of the 100K+ EESE-Pool. It updates 500 instances every quarter, and has long-term effectiveness and sustainability.
>
>    2) Governance & Consistency: We employ a **Human-in-the-loop** governance model. A dedicated maintenance team oversees the pipeline, which starts with Stratified Sampling based on discipline and difficulty distributions. This guarantees that each new iteration maintains a consistent challenge level comparable to previous versions.
>
>    3) Reproducibility: We implement a strict Snapshot Version System (e.g., EESE-v1, EESE-v2). Every released subset is permanently archived and publicly accessible, allowing researchers to reproduce historical results using specific legacy versions.
>
>    Thanks again and we will include these specific operational details in the revision.
>
>
>
> 2. **Clarity of reproducibility in response to W2**
>
>    ````
>    Heavy reliance on proprietary models for evaluation and quality control (e.g., “thinking models”) makes reproducibility questionable. This also raises a question about how users can evaluate it without access to such models.
>    ````
>
>    **Response:**
>
>    Thanks for your comments. We clarify the distinction between models used for **dataset construction** versus those required for **user evaluation**.
>
>    Regarding evaluation, proprietary models are **not** required for users to assess their own models. The released EESE benchmark comprises questions with high-quality ground-truth answers. Consequently, any user, regardless of proprietary API access, evaluates their models using our provided artifacts.
>
>    Regarding the reproducibility of the **construction pipeline**, our methodology is model-agnostic. While we utilize thinking models to ensure initial data quality, the pipeline is reproducible using high-performance open-source alternatives. We explicitly recommend these open-source alternatives to ensure the benchmark construction process remains accessible to the broader community.
>
>
>
> 3. **Clarity of model bias in response to W3 &Q2**
>
>    ````
>    W3: The use of LLMs to label and refine data raises circularity concerns and potential bias—models are used to create and test themselves.
>    Q2: How do you ensure that using LLMs to annotate or refine instances does not bias the evaluation toward their own reasoning styles?
>    ````
>
>    **Response:**
>
>    Thanks for your useful question. Our methodology mitigates bias through four distinct mechanisms:
>
>    **1) Human-Dominant Reconstruction:** Experts do not merely proofread, they fundamentally rewrite and restructure content, particularly in the Expert-Driven branch. This aggressive intervention decouples scientific concepts from the specific reasoning patterns of the assisting models.
>
>    **2) Refine and ensemble:** By injecting plausible distractors and cross-disciplinary contexts, we transform raw outputs into complex reasoning tasks. This effectively strips away the statistical patterns that models might otherwise recognize as their own. In addition, we utilize a diverse ensemble of models for drafting, preventing the benchmark from overfitting to the distribution of any single foundation model.
>
>    **3) Objective Verification:** Unlike subjective generation tasks, science questions possess deterministic ground truths. Our evaluation measures factual correctness, ensuring models are rewarded for scientific accuracy rather than stylistic mimicry.
>
>
>
> 4. **Refine of writing style in response to W4**
>
>    ````
>    The writing is verbose and could be more critical and analytical. The paper reads like an extensive benchmark manual rather than a scientific paper.
>    ````
>
>    **Response:**
>
>    We sincerely appreciate the feedback regarding the manuscript's tone and structure.  We will revise the paper for enhanced scientific inquiry. Specifically, we will move the extensive operational details of data collection and refinement rules to the Appendix to improve readability. Simultaneously, we will significantly expand the critical analysis within the manuscript, including incorporate a deeper qualitative study of model failure modes and contrast these with human reasoning patterns. Thanks again for your useful suggestions.

---

> ### Author Response · Authors · 2025-11-27
> **Official Responses to Reviewer fMyi**
>
> 5. **Analysis of failure case in response to W5**
>
>    ````
>    The evaluation mainly reports aggregate performance; there’s little qualitative analysis of what kinds of reasoning or question types models fail at.
>    ````
>
>    **Response:**
>
>    We appreciate this constructive feedback. For qualitative insight, we will incorporate a detailed failure analysis in the next version of manuscript. Here we provide a few examples:
>
>    1) A critical weakness in long-chain reasoning tasks is that **models struggle to maintain stable intermediate representations**. For instance, in mathematical challenges like the Josephus-ring variant, even advanced models fail to track state changes. Even minor index shifts cascade without self-correction, compromising the final answer.
>    2) We observe the "Reversal Curse" [1] in fill-in-the-blank tasks, where **models lack the backward-verification ability** to ensure their answers consistently align with the provided premises.
>
>
>
> 3. **Discussion of Limitations in response to W6**
>
>    ````
>    No discussion of potential biases or limitations in expert selection, data sources, or field balance.
>    ````
>
>    **Response:**
>
>    We agree that a transparent analysis of biases is essential for the credibility of any benchmark, and we will address this oversight by incorporating a dedicated "Limitations" section in the next version of manuscript.
>
>    To provide immediate clarification on our strategies: regarding **field balance**, we strictly align our data distribution with standard academic taxonomies [2] to prevent disciplinary over-representation. Regarding **data sources and expert selection**, we prioritize authoritative textbooks and enforce rigorous qualification training to minimize individual subjectivity.
>
>    Nevertheless, we openly acknowledge that intrinsic cultural or cognitive biases **remain unavoidable in any human-curated dataset**. Therefore, we position the regularly updated, open-source EESE subset not merely as an evaluation tool, but as a crucial mechanism for community auditing, allowing global researchers to identify and correct these residual biases in an ongoing, transparent manner.
>
>
>
> 7. **Analysis of refinement cost in response to Q3**
>
>    ````
>    Could you provide more transparency on the cost and human involvement behind EESE’s refinement stages?
>    ````
>
>    **Response:**
>
>    We appreciate this request and confirm the substantial scale of human and financial resources dedicated to EESE-Pool construction.
>
>    The entire construction of the 100K+ EESE-Pool involved **609 experts** and incurred a total cost of approximately **$428,057.77**, consuming roughly **30,510 expert-hours**. Compensation is strictly tiered, with core tasks like Expert-Driven Refinement (High HI) compensated at a premium rate of up to **$45 per hour**, reflecting the high professional competence required.
>
>    Comprehensive details regarding expert recruitment, qualifications, task specifications, and the tiered compensation strategy are available in the newly added **Appendix F**: Human Expert Recruitment and Involvement.
>
>
>
> 8. **Clarity of verification & leakage in response to Q4**
>
>    ````
>    Is there a plan to make the dataset partially accessible (e.g., for academic verification) without compromising leakage resistance?
>    ````
>
>    **Response:**
>
>    Thank for your question. Our"Dynamic Sampling" mechanism provides safe, representative partial access without compromising security.
>
>    Specifically, we release fully open-source 500-instance subsets that underpin the results in Table 1, with two versions already released to date. Crucially, these subsets function as statistical "slices" of the larger pool. As demonstrated in Figure 6(b), performance on the public subset **aligns rigorously with** the 100K+ EESE-Pool.
>
>    Besides, we strictly maintain these releases via a **"Snapshot Version System**", keeping every released version permanently publicly accessible. This allows the community to continuously verify the pool's quality through its evolving slices, ensuring transparency while preserving the leakage resilience that makes the benchmark valuable.

---

> ### Author Response · Authors · 2025-11-27
> **Official Responses to Reviewer fMyi**
>
> 9. **Comparison with leakage-aware benchmarks in response to Q5**
>
>    ````
>    How does EESE compare in calibration stability to recent leakage-aware benchmarks such as AntiLeakBench or LessLeakBench?
>    ````
>
>    **Response:**
>
>    Thanks for your insightful question.  Unlike *AntiLeakBench* [3] (timestamps) or *LessLeakBench* [4] (deductive filtering), EESE eliminates leakage at the source. We maintain a **private, expert-generated pool** of 100k+ instances and physically isolate it from public training corpora. By periodically sampling fresh subsets, we ensure stability by design, allowing immediate replacement of any potential leakage without relying on post-hoc detection.
>
>    Further, our results confirm this leakage-resistance. As shown in **Table 1**, top models (e.g., O3) achieve only **~40% accuracy**, lagging significantly behind human experts (**~85%**). This performance gap confirms that EESE remains uncontaminated and effectively tests reasoning capabilities.
>
>    More importantly, rigorous scientific evaluation requires **high-quality expert involvement** [5, 6]. Timestamp-based constraints are often insufficient for science, as scientific principles are timeless (a "new" date does not guarantee a novel reasoning challenge). Unlike automated filtering, EESE leverages domain experts to craft original reasoning paths. This ensures that benchmark difficulty stems from actual cognitive complexity, offering better calibration stability for assessing scientific progress.
>
>
>
> **References:**
>
> [1] Berglund et al. The Reversal Curse: LLMs trained on" A is B" fail to learn" B is A". arXiv preprint arXiv:2309.12288.
>
> [2] Standards Press. Classification and code of disciplines. 2009.
>
> [3] Wu et al. AntiLeakBench: Preventing data contamination by  automatically constructing benchmarks with updated real-world knowledge. ACL. 2025.
>
> [4] Zhou et al. Lessleak-bench: A first investigation of data leakage in llms across 83 software engineering benchmarks. arXiv preprint arXiv:2502.06215.
>
> [5] Haladyna et al. Developing and validating test items. Routledge, 2013.
>
> [6] Gierl et al. Advanced methods in automatic item generation. Routledge, 2021.

---

### Official Review · Reviewer_2rAW · 2025-11-01

**Soundness:** 1
**Presentation:** 2
**Contribution:** 1
**Rating:** 0
**Confidence:** 4

**Summary:**

This paper introduces EESE, a benchmark for scientific reasoning designed to be "leakage-resilient" by keeping 99.5% of its 100K-item dataset private (the "EESE-Pool") and releasing small, 500-item "evolving" subsets.

This review recommends a clear rejection.

The paper's "private benchmark" model is not useful to the scientific principles of reproducibility, verification, and community progress. It is fatally undermined by a direct internal contradiction, promising in its reproducibility statement to make "all relevant... datasets... publicly available" while basing its entire premise on being "non-public". Finally, it suffers from critical methodological opacity, eg, failing to provide concrete details on its human-intensive and expert-driven data creation pipeline, which makes its claim of a sustainable, "ever-evolving" benchmark untrustworthy.

**Strengths:**

The paper correctly identifies a critical and widely recognized challenge in LLM evaluation: data leakage and benchmark contamination, which can invalidate results.

The stated goal of creating a large-scale, high-quality ("Rigor") benchmark covering a wide range of scientific disciplines (500+ subfields) is ambitious and, if executed transparently, would be valuable to the community.

**Weaknesses:**

1. The paper is fatally flawed by an irreconcilable contradiction. It states its 100K-item EESE-Pool is "non-public" (Abstract, Sec 1) , yet its Reproducibility Statement (Sec 7) explicitly "guarantee[s] that all relevant... datasets will be made publicly available". This core incoherence makes the paper's premise impossible to evaluate.

2. The paper's core proposal of a non-public dataset is useless to the purpose of a scientific benchmark, which is to provide a standardized, verifiable, and public artifact for reliable comparison. This model makes replication of the paper's own results (e.g., Table 1) impossible, prevents external bias audits, and blocks community-led error correction. It functions as a private evaluation service, not a scientific contribution.

2. The paper substitutes buzzwords ("Rigor," "Data Engine") for methodology. It provides zero actionable details on its "Parallel Three-Branch Refinement Framework," (especially the later two refinement methods) failing to meet minimum reporting standards by omitting:

    - The technical or manual routing process for assigning/labeling instances for the Medium/High HI branches.
    - The recruitment, qualifications, compensation, or instructions for the "600+ experts" involved in data creation.

3. The claim of a sustainable, "ever-evolving" benchmark is unsubstantiated and appears false.

    - The process is not "low-cost." The paper's "refinement" examples (Appendix B) reveal an extremely high-expertise, human-intensive process of the problem creation, not simple editing.

   - The 100K-item pool itself is static. The paper provides no mechanism for refreshing the pool, meaning "evolving" is just bootstrap from a fixed set. This does not solve long-term benchmark staleness.

**Questions:**

- How in details are the experts instructed to label the required refinements in the middle/high HI;
- How much cost in labor/time/money is this process;
- How do you plan to make this sustainable so that you can catch up with your claim to "regularly update" the subset;
- Is there any verification mechanisms on the problems within this variety of the subjects?

**Details Of Ethics Concerns:**

The paper's Ethics Statement (Section 6) contains a verifiably false claim: it states that "no human subjects... was involved". This is directly contradicted by the paper's own methodology, which relies on the intensive labor of "600+ experts" to transcribe, expand, and refine the dataset. These experts are, by definition, human participants in a large-scale data-creation experiment. The paper fails to provide any of the required ethical disclosures for human-subjects research, such as IRB approval, participant recruitment methods, or compensation details, which is a significant violation of standard reporting practices.

---

> ### Author Response · Authors · 2025-11-27
> **Official Responses to Reviewer 2rAW**
>
> 1. **Clarity of reproducibility in response to W1**
>
>    ````
>    The paper is fatally flawed by an irreconcilable contradiction. It states its 100K-item EESE-Pool is "non-public" (Abstract, Sec 1) , yet its Reproducibility Statement (Sec 7) explicitly "guarantee[s] that all relevant... datasets will be made publicly available". This core incoherence makes the paper's premise impossible to evaluate.
>    ````
>
>    **Response:**
>    Thanks for your insightful question. We would like to clarify the scope of our data release guarantee:
>
>    The ``guarantee that all relevant datasets will be made publicly available`` in our checklist refers specifically to the **500-instance EESE subset** used to generate the results in this paper (e.g., Table 1). This subset is fully open-sourced, verifiable, and sufficient for any researcher to replicate our experiments or benchmark new models against our reported baselines. Conversely, keeping the overarching 100K+ EESE-Pool non-public is the core feature that maintains the benchmark's "Ever-Evolving" capability. Releasing the entire pool would immediately subject it to web scraping and contamination, negating the scientific contribution of this work.
>
>    Therefore, there is no contradiction in practice: the data required for **reproducibility** is public, while the data required for **longevity** is private. To eliminate any potential ambiguity, we will revise the Reproducibility Statement in the next version, ensuring the text accurately reflects this purposeful design.
>
>
>
> 2. **Clarity of misunderstanding in response to W2**
>
>    ````
>    The paper's core proposal of a non-public dataset is useless to the purpose of a scientific benchmark, which is to provide a standardized, verifiable, and public artifact for reliable comparison. This model makes replication of the paper's own results (e.g., Table 1) impossible, prevents external bias audits, and blocks community-led error correction. It functions as a private evaluation service, not a scientific contribution.
>    ````
>
>    **Response:**
>
>    Thanks for your comments. We respectfully clarify a fundamental misunderstanding regarding our reproducibility mechanism.
>
>    **The assertion that our results (e.g., Table 1) are impossible to replicate is factually incorrect**, as they are derived exclusively from the **fully open-source** 500-instance EESE subset. The specific data and code required to verify our findings are publicly available, ensuring our scientific claims are verifiable without requiring access to the withheld source pool.
>
>    Regarding external audits, the public EESE subset functions as a transparent, high-fidelity statistical sample of the larger EESE-Pool. As shown in **Figure 6(b)**, performance rankings on this public subset reliably proxy the full pool, confirming its representativeness. This allows the community to audit the benchmark's quality and bias directly via the open subset. Any errors identified by the community in these samples are used to refine the source pool, preserving the feedback loop essential for scientific rigor.
>
>    Finally, the "non-public source" structure is not a "private evaluation service," but a necessary scientific evolution to address **Data Leakage**. Since fully public benchmarks are rapidly compromised by training data contamination, withholding the source pool while periodically releasing verifiable, open-source snapshots is one of the most sustainable ways to measure true generalization. This approach balances the scientific requirement for reproducibility with the practical necessity of long-term leakage resilience.
>
>
>
> 3. **Clarity of methodology in response to W3 & Q1 &Q2**
>
>    ````
>    W3: The paper substitutes buzzwords ("Rigor," "Data Engine") for methodology. It provides zero actionable details on its "Parallel Three-Branch Refinement Framework," (especially the later two refinement methods) failing to meet minimum reporting standards by omitting:
>      - The technical or manual routing process for assigning/labeling instances for the Medium/High HI branches.
>      - The recruitment, qualifications, compensation, or instructions for the "600+ experts" involved in data creation.
>    Q1: How in details are the experts instructed to label the required refinements in the middle/high HI;
>    Q2: How much cost in labor/time/money is this process;
>    ````

---

> ### Author Response · Authors · 2025-11-27
> **Official Responses to Reviewer 2rAW**
>
> 3. **Clarity of methodology in response to W3 & Q1 &Q2  (continued)**
>
>    **Response:**
>
>    We appreciate your constructive criticism.
>
>    Technical details of the Refinement Framework (Q1): The *Parallel Three-Branch Refinement* is strictly governed by difficulty stratification. Low-difficulty instances flow to the *Enhancement by Distraction* branch, where experts generate plausible, misconception-based distractors. Medium-difficulty instances move to *Cross-Disciplinary Integration*, requiring experts to embed concepts into adjacent disciplinary scenarios. Ambiguous instances undergo *Expert-Driven Refinement*, where specialists rewrite logic chains to ensure clarity, ensuring a systematic optimization process rather than arbitrary labeling.
>
>    Regarding the workforce (Q2), the "600+ experts" comprise a high-qualification team recruited from top-tier institutions, holding a Master's or Ph.D. degree in a relevant scientific discipline. The construction spans over 3 months and involves approximately 30510 human-hours of expert labor. This substantial investment underscores why the EESE-Pool remains non-public: protecting this high-cost asset from rapid devaluation through data leakage is essential to maintaining its long-term scientific value. More details about Human Expert Recruitment and Involvement **are added in Appendix F**.
>
>
>
>
> 4. **Explain of "Ever-Evolving" Claim in response to W4**
>
>    ````
>    The claim of a sustainable, "ever-evolving" benchmark is unsubstantiated and appears false.
>      - The process is not "low-cost." The paper's "refinement" examples (Appendix B) reveal an extremely high-expertise, human-intensive process of the problem creation, not simple editing.
>      - The 100K-item pool itself is static. The paper provides no mechanism for refreshing the pool, meaning "evolving" is just bootstrap from a fixed set. This does not solve long-term benchmark staleness.
>    ````
>
>    **Response:**
>
>    We appreciate the reviewers' detailed scrutiny regarding the sustainability and cost-effectiveness of our approach.
>
>    We view the high cost of our expert-driven process as a **necessary investment in scientific rigor**, not a feasibility flaw. High-quality benchmarks require high-expertise input. By bearing this substantial "construction cost," we provide the community with a "low-cost" evaluation tool, removing the need for users to validate against massive datasets. This investment reflects our firm academic commitment to advancing evaluation standards.
>
>    Regarding the "static" pool, the dataset demonstrates strong **resilience against saturation**. Current state-of-the-art models (e.g., O3, GPT-4o) achieve only about 40% accuracy on EESE, leaving vast headroom for improvement. With our mechanism of releasing a fresh 500-instance subset every three months, the current scale ensures a dynamic evaluation landscape for a prolonged period, far exceeding the lifespan of typical benchmarks.
>
>    Finally, the "Ever-Evolving" mechanism relies on a scalable methodology, not just a fixed stockpile. We operate a fully established Data Engine pipeline (from recruitment to refinement) backed by secured human resources. Should model progress accelerate beyond expectations, we possess the validated pipeline and resolve to expand the pool further. This confirms our approach is a sustainable, long-term solution designed to evolve alongside foundation 1. models.
>
>
>
>
> 5. **Explain of  structural design in response to Q3**
>
>    ````
>    How do you plan to make this sustainable so that you can catch up with your claim to "regularly update" the subset;
>    ````
>
>    **Response:**
>
>    We appreciate the reviewers' query.
>
>    The sustainability of our quarterly updates derives directly from the structural design of the EESE-Pool. Rather than requiring labor-intensive creation from scratch for every release, our mechanism operates by **sampling from the pre-constructed, high-quality 100K+ instance reservoir**. Since the substantial initial investment in building this repository stands complete, the marginal cost and effort to validate and release each new 500-instance subset remain minimal.
>
>    Furthermore, the **operational workflows** described in our methodology, encompassing automated screening and expert verification, forms an efficient processing pipeline that enables the update process as routine maintenance. To date, we have released two versions (1,000 questions), demonstrating that our system works as designed and supports our long-term commitment to the community.

---

> ### Author Response · Authors · 2025-11-27
> **Official Responses to Reviewer 2rAW**
>
> 6. **Explain of  disciplinary diversity in response to Q4**
>
>    ````
>    Is there any verification mechanisms on the problems within this variety of the subjects?
>    ````
>
>    **Response:**
>
>    We appreciate the reviewers' query. Ensuring accuracy across such a wide variety of subjects is indeed our top priority. As outlined in **Sections 3.1 and 3.2**, we implement a multi-layered verification mechanism that specifically addresses disciplinary diversity:
>
>    1) Subfield-Matched Expert Review: To handle the broad spectrum of 5 disciplines and 500+ subfields, we distribute 600+ experts strictly based on their specific domains. For example, a question in "Medicinal Chemistry" is verified solely by experts with a background in Pharmacy or Chemistry. This "specialist-only" policy ensures domain-specific rigorousness.
>    2) Coarse-grained Quality Control: During Data Engine, we employ a suite of top-tier LLMs to flag potential logical fallacies or formatting errors. Experts then focus their efforts on verifying the scientific facts and reasoning paths of these flagged instances.
>    3) Data Refinement: We employ the Parallel Three-Branch Refinement Framework (i.e., Enhancement by Distraction, Enrichment by Cross-Disciplinary, and Expert-Driven Refinement). Eash instance is refined through one of the methods and then undergoes rigorous fine-grained quality control by human experts. This process ensures that all modifications effectively increase the challenge of the questions while steadfastly preserving their scientific accuracy.
>
> ​	We believe this rigorous, specialist-driven pipeline effectively guarantees the quality of problems across all covered subjects.
>
>
>
> 7. **In response to Details Of Ethics Concerns**
>
>    ````
>    The paper's Ethics Statement (Section 6) contains a verifiably false claim: it states that "no human subjects... was involved". This is directly contradicted by the paper's own methodology, which relies on the intensive labor of "600+ experts" to transcribe, expand, and refine the dataset. These experts are, by definition, human participants in a large-scale data-creation experiment. The paper fails to provide any of the required ethical disclosures for human-subjects research, such as IRB approval, participant recruitment methods, or compensation details, which is a significant violation of standard reporting practices.
>    ````
>
>    **Response:**
>
>    Thank you for your attention to detail.
>
>    We explicitly distinguish between "human subjects" and "expert annotators." Under standard ethical guidelines (e.g., the Common Rule [1]), human subjects research entails collecting data about individuals. In contrast, the 600+ experts in our study function strictly as content creators and QA specialists (akin to research assistants), not as experimental subjects. We evaluate LLMs, not the experts' behavioral traits. Thus, this work does not require IRB approval [1, 2] for human experimentation.
>
>    However, to address valid concerns regarding transparency, we add a detailed "Human Expert Recruitment and Involvement" section in Appendix F. This section discloses more recruitment criteria, fair compensation rates, and informed consent details, ensuring full compliance with ethical standards for expert-based data creation.
>
>
>
> **References:**
>
> [1] U.S. Department of Health & Human Services. (2018). 45 CFR 46.102.
>
> [2] Office for Human Research Protections (OHRP). (2021). Exemptions FAQ.

---

### Note · Authors · 2026-01-06

**Comment:**

Dear Chairs and Reviewers,

We are writing to formally request the withdrawal of our manuscript titled “The Ever-Evolving Science Exam” (ID: 7204), currently under consideration at ICLR2026.

After careful reconsideration, we have decided to withdraw the submission at this time. We sincerely appreciate the time and effort spent by all chairs and all reviewers on our work.

Thank you for your understanding.

Sincerely,

All auhtors of EESE

2026/01/06

**Withdrawal Confirmation:**

I have read and agree with the venue's withdrawal policy on behalf of myself and my co-authors.